# Multivoxel neurofeedback selectively modulates confidence without changing perceptual performance

Aurelio Cortese[1,2,3,4,*], Kaoru Amano[3,*], Ai Koizumi[1,3,*], Mitsuo Kawato[1,2] & Hakwan Lau[4,5]

A central controversy in metacognition studies concerns whether subjective confidence directly reflects the reliability of perceptual or cognitive processes, as suggested by normative models based on the assumption that neural computations are generally optimal. This view enjoys popularity in the computational and animal literatures, but it has also been suggested that confidence may depend on a late-stage estimation dissociable from perceptual processes. Yet, at least in humans, experimental tools have lacked the power to resolve these issues convincingly. Here, we overcome this difficulty by using the recently developed method of decoded neurofeedback (DecNef) to systematically manipulate multivoxel correlates of confidence in a frontoparietal network. Here we report that bi-directional changes in confidence do not affect perceptual accuracy. Further psychophysical analyses rule out accounts based on simple shifts in reporting strategy. Our results provide clear neuroscientific evidence for the systematic dissociation between confidence and perceptual performance, and thereby challenge current theoretical thinking.

[1] Department of Decoded Neurofeedback, ATR Computational Neuroscience Laboratories, 2-2-2 Hikaridai, Seika-cho, Soraku-gun, Kyoto 619-0288, Japan. [2] Faculty of Information Science, Nara Institute of Science and Technology, 8916-5 Takayama, Ikoma, Nara 630-0192, Japan. [3] Center for Information and Neural Networks (CiNet), NICT, 1-4 Yamadaoka, Suita City, Osaka 565-0871, Japan. [4] Department of Psychology, UCLA, Franz Hall, 502 Portola Plaza, Los Angeles, California 90095, USA. [5] Brain Research Institute, UCLA, 695 Charles E Young Dr S, Los Angeles, California 90095, USA. * These authors contributed equally to this work. Correspondence and requests for materials should be addressed to M.K. (email: kawato@atr.jp) or to H.L. (email: hakwan@gmail.com).

Confidence, the degree of certainty about our own perceptual decisions, is often considered to reflect the intriguing abilities of metacognition and introspection[1,2], which have been argued to be uniquely human[3–5]. In recent years there has been a surge of interest in how neural circuits compute confidence. Various studies, in both humans and animals, have linked specific brain mechanisms to the generation of confidence judgements[6–11] (for review, see Kepecs and Mainen[1]). Although in everyday decisions, confidence often poorly reflects decision accuracy[12,13], based on the results of laboratory behavioural studies many authors argue that confidence computation may be statistically optimal, such that confidence is directly related to the strength of the internal perceptual signal (normative views)[6,7,11,14]. Congruent with this view, researchers have also reported common neural substrates for both confidence and perceptual decisions in area LIP of macaque monkeys[7,14].

The human neuroscience literature offers a relatively mixed view. Although some studies also adopt a normative approach and define confidence in close association with the strength of the internal perceptual signal[11], several studies suggest that confidence and perceptual performance can be dissociated[9,15–19]. For example, applying transcranial magnetic stimulation (TMS) to the prefrontal cortex (PFC) resulted in a change in confidence reports without changing task performance[9]. In rodents, inactivation of the orbitofrontal cortex also impaired confidence judgements[20].

A recurrent issue in unequivocally arbitrating between these opposing views stems from experimental limitations. Confidence and perceptual performance tend to be highly correlated, so if a neural signature tracks perceptual decision accuracy, then it is not surprising that it should also reflect confidence, regardless of whether it is actually associated with the computation of confidence per se. Therefore, direct experimental manipulations for dissociating confidence from perceptual performance seem necessary[20]. However, at least in humans, such tools often lack precision. For instance, the aforementioned study with TMS to PFC only 'degraded' confidence ratings, with the ratings being less predictive of accuracy[9]. Arguably, such manipulations might have only added noise to the mechanism for the 'readout' of confidence, rather than changing the representation of confidence itself[21] (Fig. 1). Another recent study using TMS has found the opposite effect, namely that stimulation to the polar region can make confidence more predictive of accuracy[22], suggesting these effects are probably complex and not necessarily specific in direction.

Here we hypothesize that confidence is generated downstream from the processing of internal signals and perceptual decisions, and that the two processes rely on different neuronal mechanisms. As such, a systematic manipulation of confidence in precise directions (up or down) should happen independently from perceptual performance. Ideally, the change in confidence should be achieved through direct manipulation of specific brain activity patterns within prescribed regions that are independently known to reflect confidence in the same participants (see Fig. 1).

Multivoxel pattern neurofeedback, or decoded neurofeedback (DecNef), enables us to probe precisely this possibility by having participants induce brain activation patterns corresponding to a given mental state or cognitive function in specific regions[23–26]. DecNef therefore overcomes previously unresolved experimental difficulties; unlike ordinary real-time fMRI neurofeedback, which lets participants consciously self-manipulate the univariate activity level[27,28], in DecNef, multivoxel patterns of activation representing specific brain states are fed back to participants. It has been demonstrated that such reinforcement learning or neural operant conditioning of multivoxel patterns can unconsciously induce perceptual learning without stimulus presentation[23], or be used to change emotional states[24].

Before using DecNef, we first applied ordinary multivoxel pattern analysis (MVPA) to blood-oxygen-level-dependent (BOLD) fMRI data to examine the relationship between confidence and internal perceptual signal, as well as the nature of brain activation patterns related to confidence. We found correlates of confidence that seem to be independent from perceptual responses. With neurofeedback, we were then able to manipulate confidence in the expected directions (up and down), without changing perceptual performance and without the participants' awareness of the content or purpose of the manipulation. Three converging lines of evidence confirm the bi-directionality of the effects: summary statistics (individual real and expected confidence changes), standardization of DecNef effects in the second week (correction by the ratio of week-2/ week-1 effects), and mathematical modelling. Moreover, our results show that this change in confidence is unlikely to be the product of a nonspecific change in response strategy, such as a mere shift in response criterion. Thus, our results provide strong evidence for the distinct computations underlying confidence and perceptual decision making, and thereby cast doubt on currently dominant views concerning confidence and metacognition.

## Results

**Behavioural task for MVPA data acquisition.** The entire experiment consisted of six neuroimaging sessions (on separate days): retinotopy, an MVPA session and four sessions (that is, days) of DecNef (Fig. 2a). For the MVPA session, the participants performed a two-choice dot-motion discrimination task with confidence rating while in the scanner (see Fig. 2b, and 'Methods' section).

During the MVPA session, the average accuracy for motion discrimination (76.6% ± 1.5% s.e.m.) was similar to the targeted performance level (t-test against the targeted level of 75%, $t_{16} = 1.475$, $P = 0.16$, Fig. 2c). Confidence on correct trials was rated higher compared with incorrect trials (paired t-test, $t_{16} = 7.664$, $P < 10^{-6}$), in accordance with previous studies[1,21] (Fig. 2d).

**MVP analyses.** We first looked into how the three main behavioural variables, confidence (high versus low), accuracy (correct versus incorrect) and perceived motion direction (the perceptual responses, left versus right) mapped onto brain activation patterns across regions of interest (ROIs). Since confidence ratings were given on a four-point scale and the analysis performed was a binary classification, these ratings were collapsed into two classes. Thus, high and low confidence generally corresponded to ratings 3, 4 and 1, 2, respectively (see the 'Methods' section).

Specifically, we were interested in the primary visual cortex (V1/V2), motion-sensitivity areas (V3A and hMT), a key region of the ventral pathway (fusiform gyrus—FFG), and higher order processing areas previously shown to be critical in the formation of confidence judgments—inferior parietal lobule (IPL)[7,29], and lateral prefrontal cortex (lateral PFC)[8,9,18,30]. Mean decoding accuracies were averaged across all the 17 participants. Statistical analyses were performed with two-tailed t-tests against a chance accuracy of 50%. For multiple comparisons, we used the Holm–Bonferroni procedure (see the 'Methods' section), and we report corrected P values.

MVPA for correct versus incorrect trials yielded statistically significant classification accuracies in all ROIs (Fig. 3a, Supplementary Table 1). This means that task accuracy can be

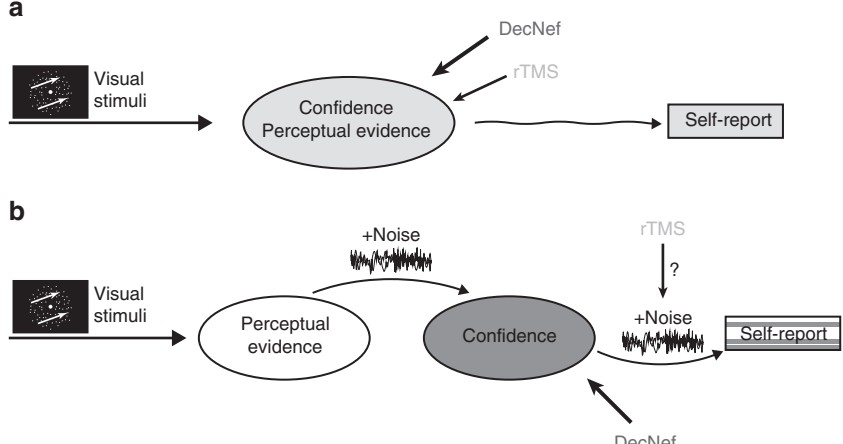

**Figure 1 | Conceptual illustration of the putative generation of confidence in the brain.** Given a certain visual stimulus as input to the system, the brain makes perceptual decisions with a corresponding level of internal perceptual evidence and noise. (**a**) Highlights the view that confidence is computed by the same neural substrates encoding perceptual evidence, and the two evolve together to give rise to a subjective report. According to this view, manipulations of confidence should change perceptual performance too. (**b**) Confidence is generated downstream from the processing of perceptual evidence, inheriting noise and signal from the earlier stages, but additional noise at this level further modulates confidence. This hierarchical view therefore considers confidence as a metacognitive process[66]. We further hypothesize that a previous study of rTMS[9] might have mainly affected the self-reporting mechanism, rather than confidence per se. This is congruent with the result that confidence ratings only became less diagnostic of accuracy, but overall confidence levels did not change in a specific direction in this previous rTMS study. If a manipulation such as decoded neurofeedback (DecNef) can specifically affect confidence representations, we should be able to selectively up- and downregulate confidence, without affecting task accuracy.

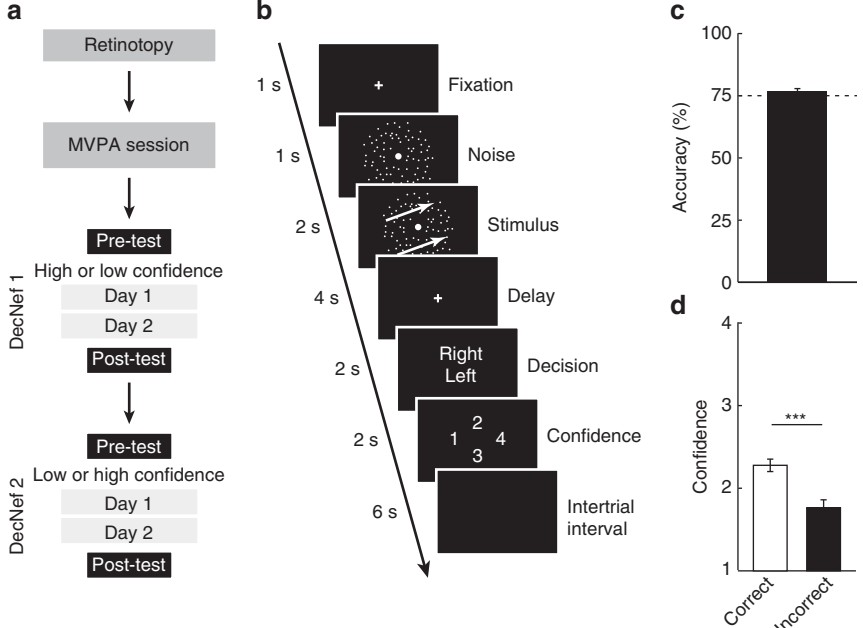

**Figure 2 | Experimental design and behavioural performance during the MVPA session.** (**a**) The experiment consisted of six neuroimaging sessions. First, we conducted a retinotopy session to functionally define the brain's visual areas. Second, we conducted an MVPA session, the data from which were used to read out the voxel-based activation patterns evoked during discrimination of motion direction with high or low confidence. Last, the multivoxel activation patterns for high or low confidence were induced in each of two separate DecNef blocks, each comprising two neurofeedback sessions (that is, 2 days), in a counterbalanced order. Each neurofeedback block was preceded by a pre-test and followed by a post-test procedure to measure the behavioural changes induced by DecNef. (**b**) The trial sequence of the two-choice discrimination task with a random dot-motion stimulus. Upon stimulus presentation, the participants were required to indicate the motion direction (leftward or rightward) and to judge confidence (four-point scale) on their perceptual decision. Importantly, the corresponding buttons were randomized and assigned after stimulus presentation, so the participants could not prepare for a specific motor response during the delay. The same trial sequence was used in the MVPA session and the pre-/post-tests. (**c**) Discrimination accuracy during the MVPA session was at a threshold level of 75% correct, achieved via stimulus titration (see the 'Methods' section). (**d**) Confidence in the correct and incorrect trials of the MVPA session. The correct trials were rated with higher confidence compared with the incorrect trials. $n = 17$, ***$P < 10^{-5}$. Centre values correspond to means, and error bars to s.e.m. DecNef, decoded fMRI neurofeedback.

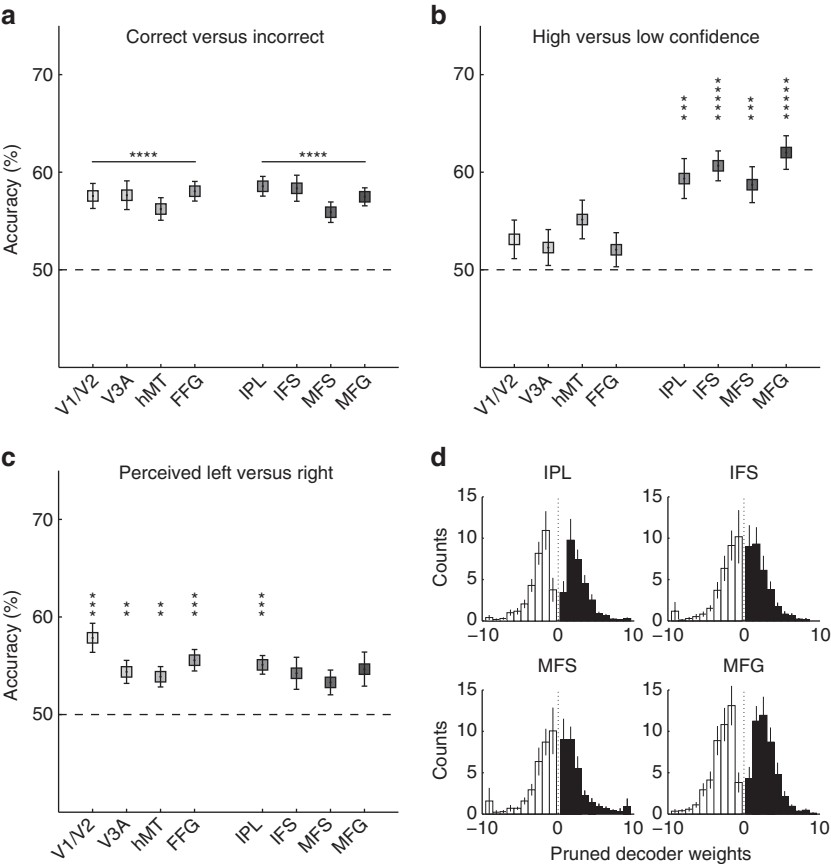

**Figure 3 | MVPA results.** MVPA results reveal that accuracy, confidence and percept can all be decoded in multiple brain regions of interest (ROIs), but the representations may differ (**a–d**). Shown here are accuracies (%) in classifying the participants' responses from their brain activation patterns within different ROIs. Accuracy in classifying (**a**) correct versus incorrect trials, (**b**) high- versus low-confidence trials and (**c**) perceived motion direction (left versus right). (**d**) Pruned weights of the confidence decoder. The SLR algorithm automatically selected relevant voxels that carried information to decode confidence. The weights were distributed to both the negative and positive sides of the abscissa; thus, the classification of high versus low confidence depended upon spatial activation patterns, rather than general activation changes; $n = 17$, $*P < 0.05$, $**P < 0.01$, $***P < 0.005$, $****P < 10^{-3}$, $*****P < 10^{-4}$; $P$ values corrected for multiple comparisons (Holm–Bonferroni). Centre values correspond to means, and error bars to s.e.m. FFG, fusiform gyrus; IFS, inferior frontal sulcus; IPL, inferior parietal lobule; MFG, middle frontal gyrus; MFS, middle frontal sulcus.

reflected by the spatial pattern of fMRI activity at different stages of the visual and cognitive processing hierarchy.

Successful MVPA for confidence (Fig. 3b, Supplementary Table 2), conversely, was associated mostly with frontoparietal areas (IPL, and lateral PFC subregions), although hMT marginally contained fMRI signals that allowed the prediction of the confidence level.

Perceived motion direction (Fig. 3c, Supplementary Table 3) could be also discriminated at above chance levels from the voxel activation patterns in multiple ROIs in the visual processing areas as well as the parietal region.

One concern is that the successful classification of confidence may solely reflect the difference in the overall activation level between the two behavioural conditions (that is, high and low confidence), rather than fine-grained spatial patterns of activity. Yet, this was unlikely, as the weights of the sparse logistic regression (SLR) classifier in each ROI were symmetrically distributed around zero, having both negative and positive values (Fig. 3d). If the high decoding accuracy of confidence were solely due to a general increase or decrease in the BOLD signal, we would expect the SLR weights to be skewed toward positive or negative values, respectively. Thus, the confidence MVPA results reflect true spatial patterns of voxel activations.

Another concern is that high classification accuracy in some ROIs, for example, for confidence in the frontal areas, may simply

reflect the larger number of voxels relative to other regions. To control for this confound, we repeated the same MVP analyses while equating the number of voxels across all ROIs. Voxels were selected based on their $t$-value in a univariate general linear model (GLM) analysis contrasting stimulus presentation versus a blank baseline. Even after controlling for the voxel number, results remained qualitatively very similar (Supplementary Fig. 2a–d).

To further address the question of how confidence judgments may emerge, we looked into the relationship between the internal sensory signal, and task accuracy or confidence. Previous work suggests that confidence is a direct transformation of the internal sensory signal driving perceptual decisions[1,6,7,11,14]. To examine this hypothesis, we first trained a classifier to distinguish between perceived leftward versus rightward motion. We then rectified the output decision variable of the classifier (that is, the linear discriminant function, LDF) by taking absolute values, and used that rectified signal to predict task accuracy. In other words, if the classifier considers the leftward or rightward motion signal to be of extreme magnitude, we expect task accuracy to be high, just as signal detection theory predicts[31] (Fig. 4a,b). Indeed, as expected, a direct classification without rectification resulted in chance performance of the classifying algorithm (Fig. 4c). However, upon rectification, the information on the magnitude of the motion contained in the voxels' activation values was successfully utilized by the classifier to discriminate between correct versus incorrect

trials (Fig. 4d, Supplementary Table 4). The distributions of rectified LDF output values for each ROI, all participants pooled (Fig. 4e), qualitatively resembled the mean differences between correct and incorrect trials and were reminiscent of the expected pattern for a good separability (Fig. 4b-right), constituting the positive results reported above (Fig. 4d, Supplementary Table 4).

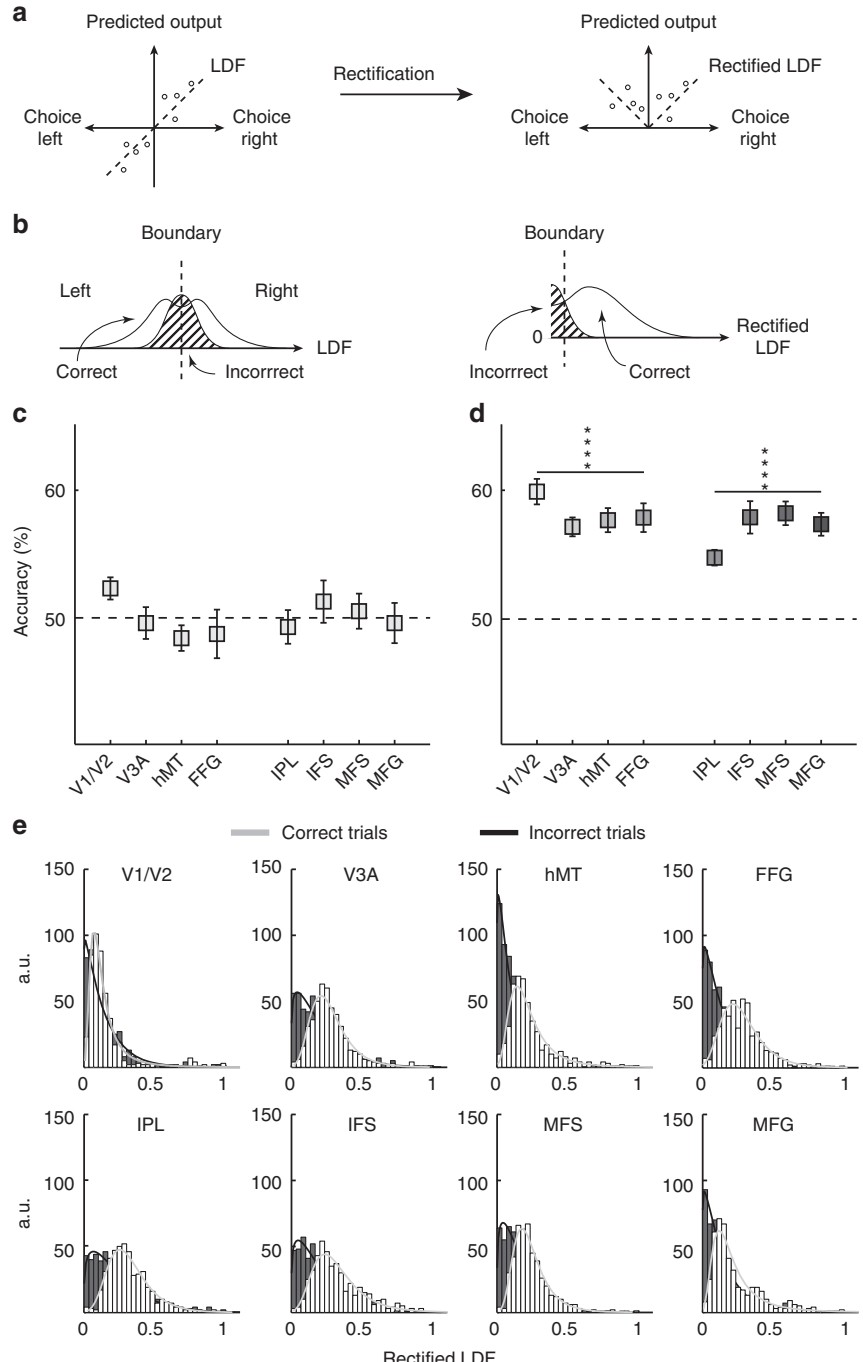

**Figure 4 | Rectification analysis for task accuracy.** (**a**) We initially constructed a decoder that classified perceptual responses into leftward versus rightward motion. When generalizing to correct versus incorrect trials, the output of this decoder (the linear discriminant function, LDF) was rectified (absolute value taken where zero was the discrimination criterion for left versus right motion) before assigning labels to each trial. (**b**) Rationale of the analysis: before rectification, the output of the LDF does not distinguish between correct and incorrect trials. The distribution of LDF output values for the incorrect trials is expected to be unimodal and centred at the left–right decision boundary, while the distribution for correct trials is expected to be bimodal, with peaks distributed on either side of the left–right decision boundary. Importantly, the two distributions should have approximately the same mean. However, upon rectification, the distribution for correct trials will have a higher mean value and therefore a new boundary can be set to discriminate between correct versus incorrect trials. (Please see the main text for a more detailed explanation). (**c**) As expected, before rectification, the decoder for perceptual responses fails to generalize to the discrimination of correct versus incorrect trials. (**d**) Following rectification, the information contained in the voxels' spatial activation patterns can be used to successfully discriminate between correct versus incorrect trials. (**e**) Distributions of normalized rectified LDF values for correct and incorrect trials (all participants pooled; a.u, arbitrary unit); $n = 17$, ****$P < 10^{-3}$; $P$ values corrected for multiple comparisons (Holm–Bonferroni). Centre values correspond to means, and error bars to s.e.m. ROI labels as in Fig. 3.

Again, as expected, direct application of the LDF output values from the perceived motion-direction classification to decode confidence resulted in only chance-level performance (Fig. 5a). However, in sharp contrast to the application of the rectified LDF values to perceptual accuracy (as shown in Fig. 4), even after rectification, the LDF values from the perceived motion decoding still could not predict confidence in any ROI after correction for multiple comparisons (Fig. 5b). At uncorrected values, hMT and FFG were slightly significantly different from chance (Supplementary Table 5; $P = 0.0393$ and $P = 0.0423$, respectively). Thus, somewhat surprisingly, applying the same transformation of the LDF trained on the perceived motion direction resulted in much poorer decoding of confidence than with decoding of perceptual accuracy. The distributions of rectified LDF output values were indeed almost identical between high- and low-confidence trials (Fig. 5c), which explains why it was impossible to achieve above-chance classification of confidence in most ROIs (Fig. 5b).

Nevertheless, classification is a complicated process and the above analysis may lack transparency. Therefore, we also adopted a simpler approach, to graphically assess the relationship between confidence and the output of the linear classifier obtained from

training on leftward versus rightward motion. If the LDF and confidence are consistently related, one should see a specific pattern of association between the two. However, we found that this was not the case for most ROIs except for V1/V2 (Fig. 6, Supplementary Figs 3,4), thereby confirming the conclusions of the previous analyses.

The results from the decoding analyses for sensory signal, task accuracy and confidence, as well as their relationships, suggest that confidence may involve a distinct late-stage process downstream of perceptual decisions. Furthermore, the neural representation of confidence seems to be independent from task accuracy and the percept, as demonstrated by the differential results in MVPA, specifically between confidence and accuracy, in the output of the rectification analysis. Taken together, these results set the stage for the use of decoded neurofeedback to directly test the hypothesis that confidence is generated downstream of and distinctly from the processes for perceptual decisions. If the hypothesis reflects the actual confidence computations in the brain, we should see an effect of DecNef restricted to the confidence dimension, without introducing any change in task accuracy (Fig. 1).

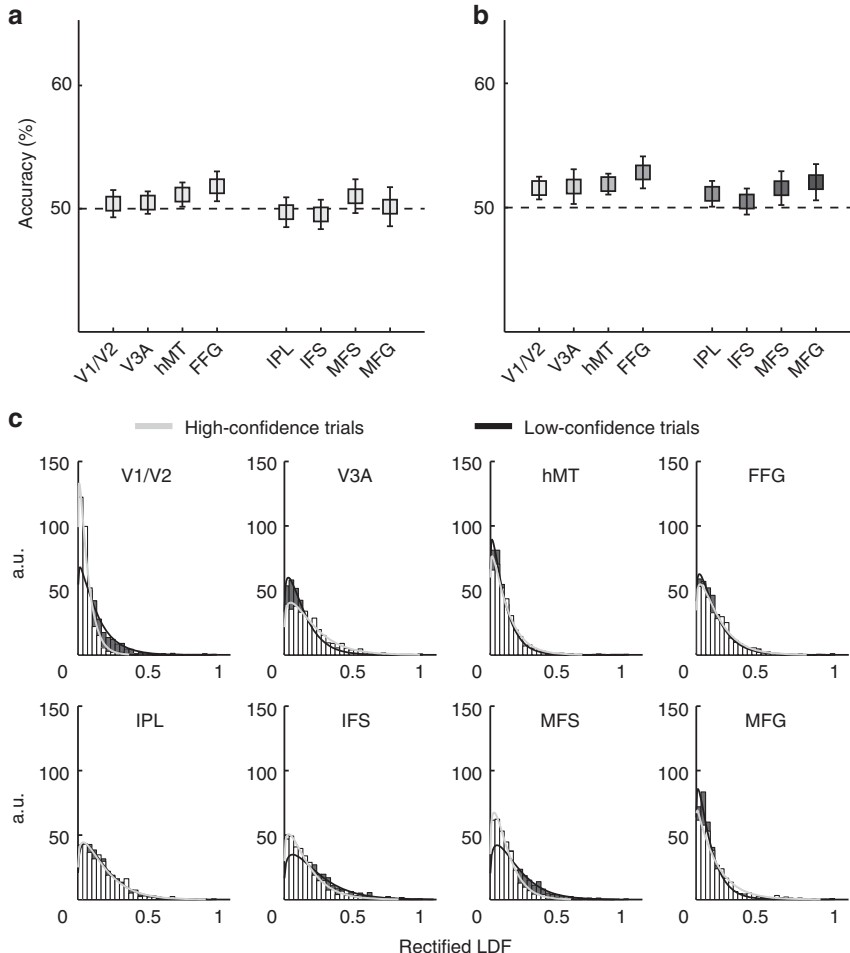

**Figure 5 | Confidence is not just the rectified perceptual response.** We applied the same steps described in Fig. 4a,b, and found that the rectified perceptual response cannot predict confidence in most brain regions. (**a**) As expected, before rectification, the decoder for perceptual responses cannot generalize to confidence. (**b**) If confidence were a direct product of the transformed internal sensory signal, this would predict that by rectifying the output of the decoder, confidence levels could be classified. Our results suggest that this was not the case for most ROIs, as only a marginal improvement of decoding accuracy was found in hMT and FFG—which did not survive correction for multiple comparisons. (**c**) Distributions of normalized rectified LDF values for high- and low-confidence trials are highly similar (all participants pooled; a.u., arbitrary unit), confirming the negative results in **b**; $n = 17$, centre values correspond to means, and error bars to s.e.m. ROI labels as in Fig. 3.

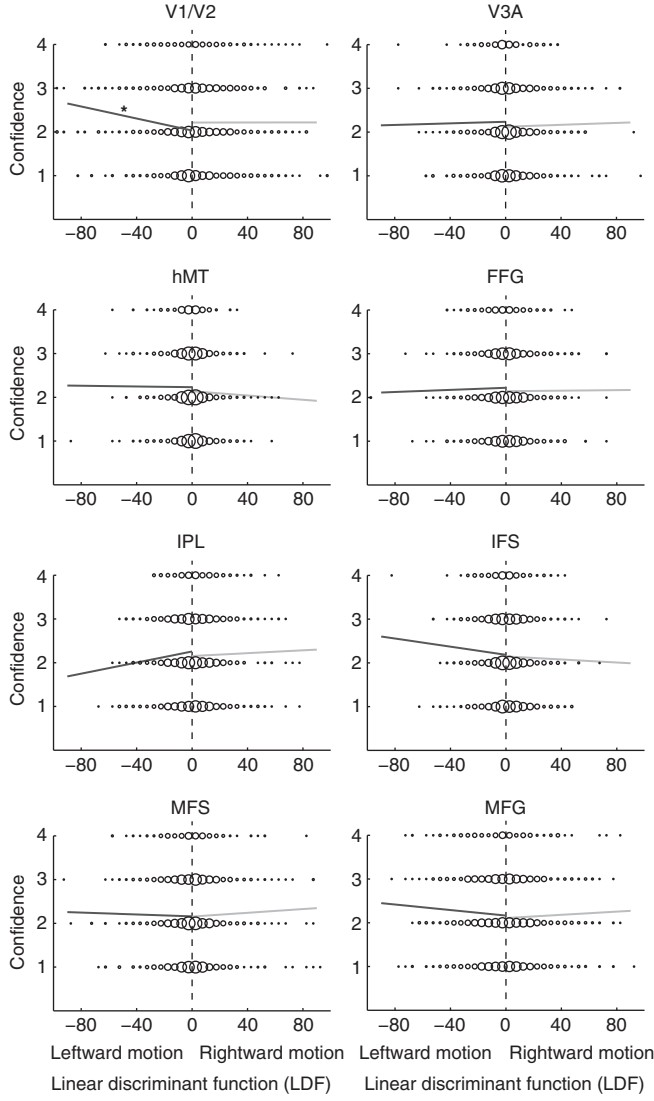

**Figure 6 | Relationship between linear classifier output and confidence.**
Here we graphically assessed the relationship between confidence and the output of the classifier (linear discriminant function, LDF) constructed on the basis of the perceived direction of motion (leftward, rightward). Larger magnitude of LDF value represents trials of higher signal strength. For each ROI, black circles represent binned data points pooled from all the participants, at each confidence level, respectively. The size of the circles reflects the number of data points within each bin; each side of the LDF function was subdivided into 20 bins. Thick lines (dark for negative LDF values, light for positive LDF values) are linear fits to the LDF against confidence levels. On the basis of normative optimality models, one would expect higher absolute LDF magnitude to be associated with higher confidence ratings, thereby forming a 'v-shaped' pattern on these plots. V1/V2 alone shows a relevant significant correlation between negative LDF values (leftward motion), and confidence (Pearson's $r$, corrected for multiple comparisons across ROIs). In all other ROIs, there seem to be negligible meaningful relationship between LDF magnitude and confidence. $*P < 0.05$ ROI labels as in Fig. 3.

**Basic findings from the neurofeedback experiment.** After construction of a decoder for confidence in the MVPA session (that is, a decoder that can reliably classify high versus low confidence from brain activation patterns), participants were screened out either if they could not or declined to come back for the DecNef sessions, or due to low decoding accuracy for confidence in

frontoparietal ROIs. Ten participants thus proceeded through the neurofeedback training. The selection was necessary because our previous studies[23,24,26] show that for DecNef to induce significant multivoxel activation changes, initial decoding accuracy has to be sufficiently high. In DecNef, multivoxel activation changes are the experimental independent variables, and if they cannot be produced, experiments cannot take place. Each participant participated in four neurofeedback sessions (that is, days), two consecutive sessions each for high- and low-confidence inductions, with the order of high- and low-confidence inductions counterbalanced across participants. Both high-confidence DecNef sessions and low-confidence DecNef sessions were preceded and followed by behavioural pre-test and post-test (Fig. 2a).

For all induction trials in the fMRI scanner, the participants were instructed to manipulate and change their brain activity to enlarge the feedback disc presented at the end of each trial as much as possible. The size of the disk represented the monetary reward they would receive on each trial (Fig. 7a). Because dot-motion stimuli were presented during DecNef, we hypothesized that subsequent presentation of such stimuli in the post-test may recall the DecNef-induced brain activity patterns via associative learning[26,32]. Throughout the experiment, the participants were naive with regard to the purpose of the neurofeedback sessions. As in a previous study[23], even when asked in a forced-choice manner which experimental group they thought they belonged to (whether they first did high-confidence DecNef or low-confidence DecNef, followed by the other condition), the participants as a group could answer only at chance (five correct, four incorrect, one not applicable, Chi-square test, $\chi^2 = 0.225$, $P = 0.64$). Furthermore, participants' reported strategies revealed an absence of knowledge regarding the purpose of the inductions in the DecNef sessions (see Supplementary Table 6).

Figure 7b shows confidence ratings at the four timings averaged across participants for the two groups (high- then low-, or low- then high-confidence DecNef). We emphasize that both groups of participants had almost identical mean ($\pm$ s.e.m.) starting confidence levels (Fig. 7b; group D-U, confidence at t1: $\bar{C} = 2.011 \pm 0.132$; group U-D, confidence at t1: $\bar{C} = 2.006 \pm 0.130$; paired $t$-test $t_4 = 0.023$, $P = 0.98$). A closer examination of individual data reveal that the results at the group level (Supplementary Fig. 5a) were mirrored at the individual level (Supplementary Fig. 5b). Moreover, confidence changes had a clear common trend across participants well manifested when data were realigned to a common starting point (Supplementary Fig. 5c). Most importantly, DecNef induced bi-directional changes in confidence ratings as measured in the post-tests. That is, the participants were more confident than before in their perceptual decisions after the high-confidence DecNef session, and showed decreased confidence following the low-confidence DecNef session, as evident from Fig. 7b and Supplementary Fig. 5. Three independent lines of evidence as detailed below statistically supported this observation.

**Neurofeedback induced significant confidence changes.** Qualitatively, order and type of DecNef (high- then low-, or low- then high-) seemed to have had a large influence on how confidence was experimentally manipulated (Fig. 7b). Specifically, data suggest that 7/10 cases in the low-confidence DecNef and 9/10 cases in the high-confidence DecNef showed changes in the expected directions (Fig. 7c, Supplementary Fig. 7c). Supposing, as a null hypothesis, that the direction of each confidence change occurs at random, the associated probability is then 1/2. Assuming that each DecNef session is independent, the cumulative binomial probability to obtain 16/20 matches is

$P(X > 15) = 0.0059$. This null hypothesis that after each DecNef session increase or decrease in confidence occurred at random is thus statistically implausible. Furthermore, as 16/20 matches were observed, confidence changed in the expected directions as designed in the experiment, contrary to the prediction of the null hypothesis.

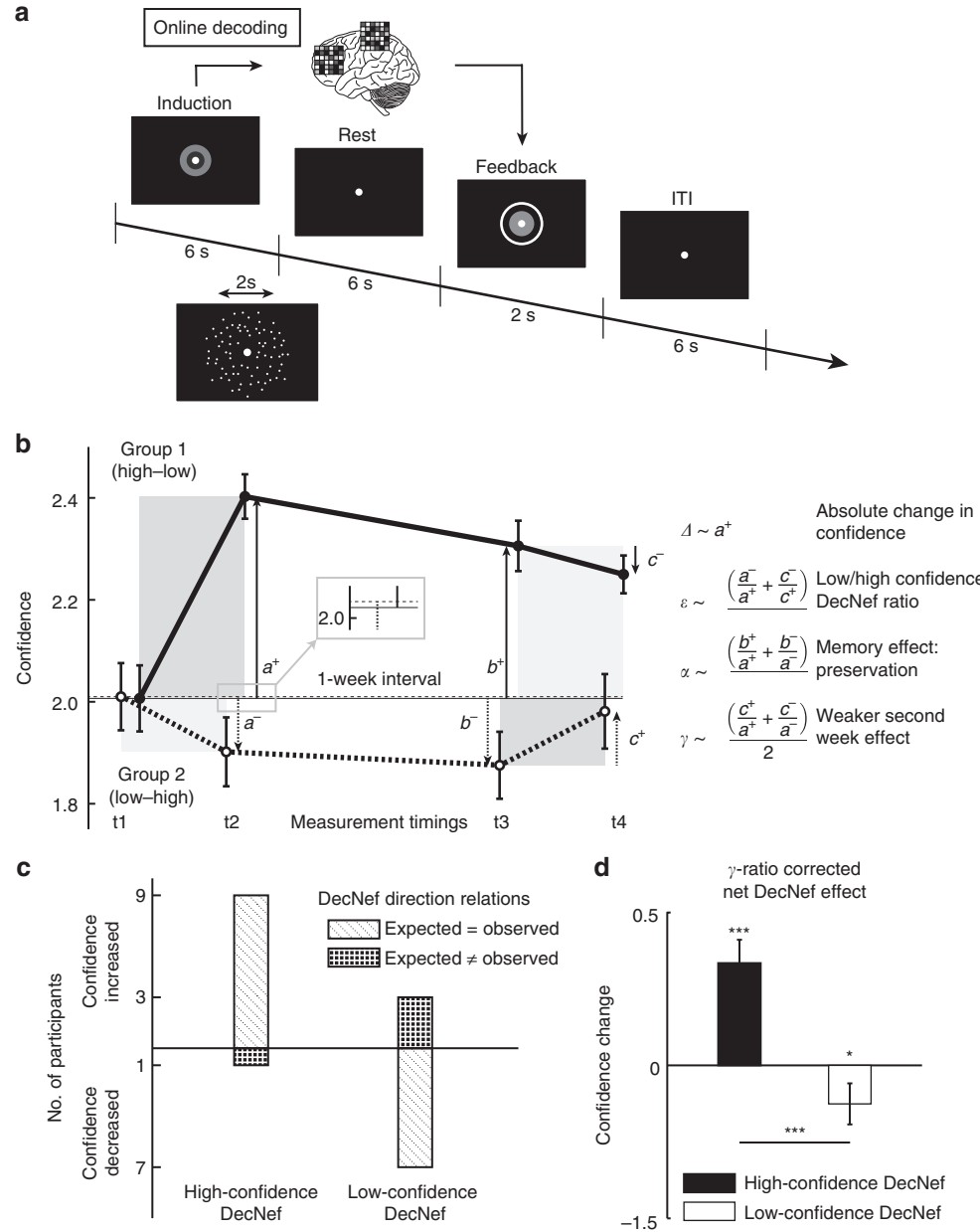

**Figure 7 | DecNef trial design and bi-directional confidence manipulation. (a)** The sequence of a single DecNef trial. The participants were instructed to modulate their brain activity during the induction period with the goal of making the grey feedback disc as large as possible. Online decoding was performed with fMRI BOLD activity patterns from all four frontoparietal ROIs (IPL, IFS, MFS and MFG) during 6–12 s after the onset of the induction cue, to account for the hemodynamic delay. The feedback disc size indicated the amount of monetary reward earned in that trial (max = 18.75 yen, approximately 0.15 US dollars in each trial), and was determined by the likelihood of the real-time activation patterns being classified as high or low confidence (depending on the session), given previous subject-specific MVPA results. **(b)** Confidence measurements for the two groups across four time points of the experiments: the pre-, post-tests on week one (1 and 2, respectively), and pre- and post-tests on week 2 (3 and 4, respectively). Thick solid line shows the results for high–low confidence DecNef order group, thick dashed for low–high confidence DecNef order group. Grey backgrounds (darker for high confidence DecNef) indicate when DecNef training took place. Vertical arrows $a$, $b$ denote the change in confidence from time point 1 while the line $c$ represents the difference from the previous time point 3. Superscripts $+$ and $-$ indicate the directionality of the manipulations: high and low confidence DecNef, respectively. The ratio formulations of the raw parameters are shown on the right side; $n = 5$, centre values correspond to means, and error bars represent s.e.m. **(c)** Illustration of the binomial probability between the observed and expected directions of confidence changes. Nine out of 10 changes were in the expected direction for high confidence DecNef, and 7/10 for low confidence DecNef. **(d)** Bi-directional net effects of high- and low-confidence DecNef. The second week confidence change was standardized with the gamma ratio week-2/week-1 effects (indicated in **b**), to account for the interference of the first week DecNef with the second week effects. The changes were significant in both the directions, demonstrating that confidence was bi-directionally manipulated by neurofeedback training, $n = 10$, centre values correspond to means, and error bars represent s.e.m. ***$P < 0.005$, *$P < 0.05$.

Quantitatively, to analyse DecNef effects on confidence, we utilized a mixed-effects repeated-measures analysis of variance (ANOVA; with between-subjects factor of group (low–high and high–low orders) and within-subjects factor of timing (Fig. 7b, timing 1, 2, 3 and 4)). The ANOVA analysis resulted in a strongly significant interaction between factors group and timing, $F_{3,24} = 8.650$, $P < 0.001$ (univariate effect). Main effects of factors timing, $F_{3,24} = 2.555$, $P = 0.079$, and group, $F_{1,8} = 3.674$, $P = 0.092$, were close to significance.

From Fig. 7b, as a qualitative assessment of DecNef effects on confidence, it is apparent that high-confidence DecNef had a larger influence than low-confidence DecNef, and the changes in confidence induced by DecNef were larger in the first week than those in the second week irrespective of high- or low-confidence DecNef. In addition, once a new confidence level had been acquired, it was seemingly maintained to some extent over the weeklong interval; this in turn strongly interacted with any subsequent DecNef training (see Supplementary Fig. 5). By simply utilizing the ratios of the confidence changes between different time points and directions, we can quantify these main consequences of bi-directional DecNef. The derived parameters were the initial (first week) mean absolute change in confidence by high-confidence DecNef ($\Delta \sim 0.396$), the smaller and with opposite sign low-confidence DecNef effect ($\varepsilon \sim -0.400$), the preservation of changes between DecNef sessions across the weeklong interval ($\alpha \sim 0.997$), and the reduced second-week effect ($\gamma \sim 0.390$) (Fig. 7b, right side).

Owing to the weaker DecNef effects in the second week and group orders, the true DecNef effects would be partly masked if we had simply averaged the first-week and second-week effects (raw confidence changes for high- and low-confidence DecNef, Supplementary Fig. 6a). To avoid this masking, the ratio of the second-week DecNef with the first-week effect ($\gamma$) is taken into account, and we multiplied the second-week individual confidence changes by $1/\gamma$. The true DecNef effects transformed as the first-week effects can then be computed as averages from all 10 DecNef sessions for both high and low confidence. The resulting changes in confidence by high- and low-confidence DecNef were both significantly different from zero, and this provided the second piece of evidence for the bi-directionality of DecNef effects (Fig. 7d; one-sided $t$-test: increase for high-confidence DecNef, $t_9 = 4.39$, $P = 0.0009$; decrease for low-confidence DecNef, $t_9 = -1.88$, $P = 0.0467$; high—higher than low-confidence DecNef, $t_9 = 4.314$, $P = 0.001$). Critically, although the raw confidence values (Supplementary Fig. 6) show the same trend, the corrected values, after taking into account session, time and learning effects, show a clearer pattern.

Finally, for a thorough analysis of these effects, we applied nonlinear modelling to formally decompose and characterize the main behavioural consequences of DecNef, accounting for both high- and low-confidence neurofeedback, as initially explored above and illustrated in Fig. 7b. Specifically, we fitted a system of nonlinear parametric equations with four global parameters $\Delta$, $\varepsilon$, $\alpha$, $\gamma$; same as in Fig. 7b. Additional models are described in Supplementary Fig. 7a, Supplementary Table 7 and Supplementary Note 1. To compare the models, we used the corrected Akaike Information Criterion (AICc, see the 'Methods' section)[33,34]. Model parameters were estimated under least square minimization and through model averaging, given model selection uncertainty with three models having $\Delta$AICc$<2$ (ref. 34) (Supplementary Fig. 7a,b, Supplementary Table 7). The three best models possessed non-zero and negative $\varepsilon$, thus the nonlinear modelling results clearly demonstrated that not only high-confidence DecNef but also low-confidence DecNef induced confidence change in the expected directions. Consequently, AICc further supported the bi-directionality of DecNef effects

(Supplementary Fig. 7a,b). The estimated delta parameter ($\Delta$) was 0.37; thus high-confidence DecNef on the first week increased confidence by 0.37, about 20% change in confidence. Alpha ($\alpha$) was 0.83; hence on average only 17% of the first week effect was lost during the one-week interval owing to memory decay. Epsilon ($\varepsilon$) was $-0.35$, thus the low-confidence DecNef effect was opposite in its sign and 35% of the magnitude of that of the high-confidence DecNef. Gamma ($\gamma$) was 0.19, and thus the second week effect was only 19% of that of the first week (Supplementary Fig. 7b). Overall, supporting the validity of the estimated model parameters, a simple correlation analysis between observed confidence changes and estimated changes computed with the mathematical model equations, yielded a highly significant correlation ($n = 20$, Pearson's $r = 0.748$, $P < 10^{-5}$, Supplementary Fig. 7c).

**The specificity of neurofeedback effects on confidence.** Similar to the analysis of Fig. 7d, to corroborate the previous result using the $\gamma$-ratio to account for the second-week weaker effect of DecNef and identify the mean true effect, individual mean confidence change values in the second week were multiplied by $1/\gamma$ ($\gamma$ parameter obtained through model averaging). Significant effects for the two DecNef trainings were confirmed (one-sided $t$-test, $t_9 = 3.64$; $P = 0.0027$, $t_9 = -1.92$, $P = 0.0436$, for high- and low-confidence DecNef, respectively; $t_9 = 3.486$, $P = 0.0034$, for high- versus low-confidence DecNef; Supplementary Fig. 7d). This result, together with the negative and non-zero value of the estimated $\varepsilon$ parameter provided the third line of evidence for the bi-directionality of DecNef-induced confidence changes. Furthermore, it emerges that a more sophisticated method used to evaluate net confidence changes resulted in larger DecNef effects being detected. Indeed, by comparing Fig. 7d and Supplementary Figs 6 and 7, this qualitative trend can be directly appreciated, supporting the modelling results. Therefore, although the effects of DecNef were complex, as they interact with session order and time, there is strong evidence to support that it significantly modulated confidence in both the positive and negative directions.

Critically, task accuracy in the two-choice discrimination task did not change between the pre- and post-tests; this rules out that confidence changes were simply due to a change in discrimination accuracy, which would have trivialized our finding as confidence and accuracy are typically confounded[21,35]. A two-way repeated-measures ANOVA (factors of neurofeedback and time) showed nonsignificant interaction ($F_{1,9} = 0.030$, $P = 0.867$) and non-significant main effects of time ($F_{1,9} = 0$, $P = 0.994$) and neurofeedback ($F_{1,9} = 1.854$, $P = 0.206$; Fig. 8a).

Interestingly, the effect of DecNef on confidence was more robust for incorrect than correct trials (Fig. 8b,c, data reported were corrected for the second-week effect with the $\gamma$-parameter as in the previous results for general confidence changes. Uncorrected data are reported in Supplementary Fig. 6). The true change in confidence for incorrect trials in high-confidence DecNef was significantly different from 0 (one-sided $t$-test for increase, $t_9 = 3.349$, $P = 0.0043$, Fig. 8b), and it was close to significance in low-confidence DecNef (one-sided $t$-test for decrease, $t_9 = -1.328$, $P = 0.108$, Fig. 8b). The changes were not significant for correct trials in both DecNef type of training (Fig. 8b). A three-way repeated-measures ANOVA (factors of response accuracy (correct versus incorrect), neurofeedback (high- versus low-confidence DecNef) and time (pre- versus post-test), see Fig. 8c) resulted in a close-to-significance three-way interaction ($F_{1,9} = 4.166$, $P = 0.072$), indicating that confidence seemed to change asymmetrically for correct and incorrect responses. This was supported by the significant main effect of response accuracy ($F_{1,9} = 23.945$, $P = 0.001$). Main effect of time was also significant ($F_{1,9} = 7.419$, $P = 0.023$). The interactions

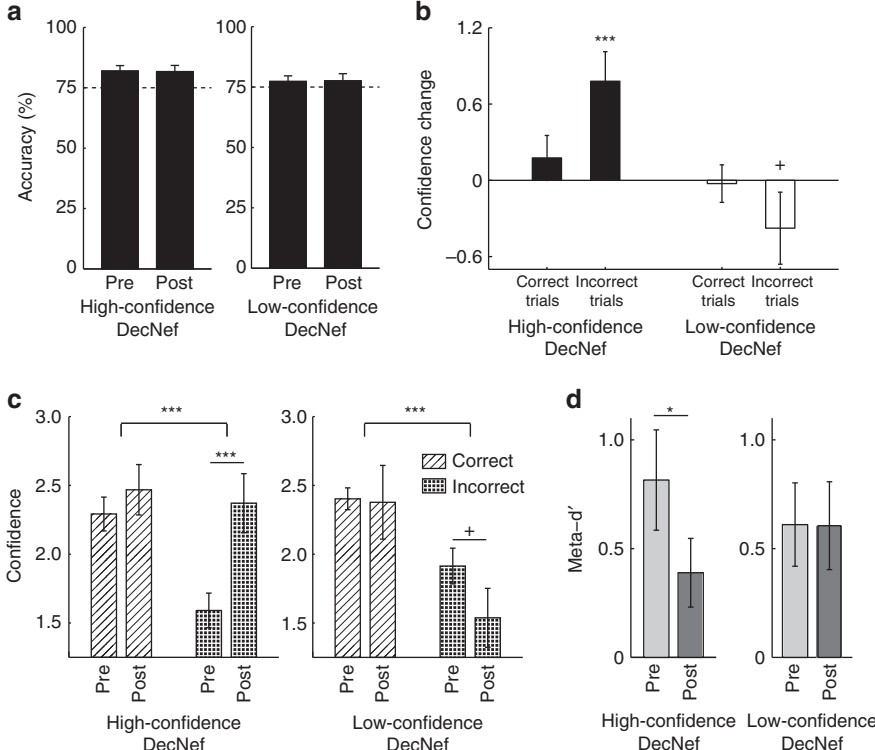

**Figure 8 | Additional behavioural effects of DecNef.** (**a**) Discrimination accuracy in pre- and post-tests did not change; a two-way ANOVA with repeated measures resulted in a nonsignificant interaction, as well as nonsignificant main effects of time and neurofeedback. (**b**) Net confidence changes for correct and incorrect trials, in high- and low-confidence DecNef. The data plotted take into account the order and interference of DecNef sessions and integrate the estimated gamma parameter (see main text). Qualitatively, DecNef had a larger effect on incorrect trials. Confidence change for incorrect trials in high-confidence DecNef was significantly different from 0, while in low-confidence DecNef the change was close to significance. A clear trend emerges from these data, indicating that the effect had opposite effects in high- and low-confidence DecNef, and that it was larger for incorrect trials. (**c**) Asymmetrical changes in confidence for correct and incorrect trials, in high- and low-confidence DecNef, plotted with respect to pre- and post-test measures. A three-way ANOVA (factors of response type, neurofeedback and time) with repeated measures showed a close-to-significance three-way interaction, a significant interaction between neurofeedback and time, and a significant main effect of response accuracy. As in the previous panel, the data plotted take into account the order and interference of DecNef sessions and integrate the estimated gamma parameter. (**d**) Meta-d' was significantly reduced following high-confidence DecNef, indicating that a mere change in criterion cannot account for the results reported in **b,c** and Fig. 7d; see main text for explanation. Two-way ANOVA with repeated measures showed no significant interaction, but the main effect of time was close to significance; $n = 10$, $^+P = 0.108$, $^*P < 0.05$, $^{***}P < 0.005$. Centre values correspond to means, and error bars to s.e.m.

between response accuracy and neurofeedback ($F_{1,9} = 1.391$, $P = 0.269$) as well as between response accuracy and time ($F_{1,9} = 0.445$, $P = 0.521$) were not significant. Reflecting the result of the mixed ANOVA reported for Fig. 7b, the interaction between neurofeedback and response accuracy was also significant ($F_{1,9} = 12.539$, $P = 0.006$). Reflecting these outcomes, the fractions of single confidence levels in correct and incorrect trials were not equally affected by DecNef (Supplementary Fig. 8a,b). A possible interpretation is that confidence associated with incorrect responses is more malleable, owing to a weaker and noisier perceptual signal.

The result that DecNef influenced confidence more markedly in incorrect trials seems to suggest that the effect was more complex than a mere change in responding criterion or strategy; if participants generally tended to report higher confidence after high-confidence DecNef, this should be reflected as a more uniform effect in both correct and incorrect trials. However, one may also worry that small changes in confidence on correct trials may be due to a ceiling or floor effect. To formally test whether the effect of DecNef can be instead explained in terms of a criterion shift in signal detection theoretic (SDT) terms[31], we performed a formal type-2 analysis[35]. Using the raw confidence ratings in both correct and incorrect trials, we computed meta-d' (ref. 36), a type-2 SDT measure of metacognitive sensitivity that

represents how well confidence reflected accuracy over trials. Given that meta-d' measures are by nature independent of criterion changes (both type-1 and type-2 (ref. 36)), a mere change in criterion should leave meta-d' constant. We found that meta-d' decreased following high-confidence DecNef (Fig. 8d). Such a change was not seen following low-confidence DecNef, and a two-way ANOVA with repeated measures (factors of neurofeedback and time) showed no significant interaction ($F_{1,9} = 2.822$, $P = 0.127$), albeit the factor time was close to significance ($F_{1,9} = 3.325$, $P = 0.102$), while neurofeedback ($F_{1,9} = 0.002$, $P = 0.968$) was not significant. *Post hoc* two-tailed paired *t*-tests to contrast pre- and post-tests displayed a significant effect of high-confidence DecNef ($t_9 = 2.912$, $P = 0.0173$), while not for low-confidence DecNef ($t_9 = 0.026$, $P = 0.980$) on meta-d'. These results suggest that at least for high-confidence DecNef, the behavioural effect cannot be accounted for by a simple change in criterion according to a signal detection theoretic model; the effect was specific for incorrect trials.

**The mechanisms and factors leading to neurofeedback success.** To verify that DecNef really had a causal role in inducing the subsequent behavioural changes, we looked into how well performance in neurofeedback training (that is, likelihood of

successful induction of the targeted brain activity patterns) correlated with changes in confidence. Because there were in total four DecNef sessions, given that each of high- and low-confidence DecNef consisted of two neurofeedback sessions, we looked separately at the effect of induction success for the first and second day of DecNef for each condition. We found that induction success on the second day of the neurofeedback training for each DecNef condition (high and low confidence), that is, the session immediately preceding post-test, strongly correlated with the change in confidence (Fig. 9a, black line, $n = 20$, Pearson's $r = 0.680$, $P = 0.00096$; where $n$ indicates the total number of DecNef conditions tested, that is, 10 participants at two conditions each, high- and low-confidence DecNef). Specifically, this effect was mainly driven by the high-confidence DecNef condition (Fig. 9a, dark grey line, $n = 10$, Pearson's $r = 0.703$, $P = 0.023$); the correlation was not statistically significant for low-confidence DecNef alone (Fig. 9a, light grey line, $n = 10$, Pearson's $r = 0.083$, $P = 0.820$).

We also analysed the relationship between change in confidence and induction success in the first day of DecNef in each condition (high and low confidence), and found weaker correlations (see the 'Methods' section, and Supplementary Fig. 9). Given that induction success was slightly higher in the second day for all conditions averaged (Supplementary Fig. 10a), these results are perhaps unsurprising because post-test followed immediately the second day rather than the first day of DecNef in each condition. Furthermore, precluding the possibility that the influence of neurofeedback training on confidence may have simply reflected the positive effect associated with monetary reward and/or correct performance, the total amounts of monetary reward for high-

confidence and low-confidence induction were not statistically different (2-days average ( ± s.d.) monetary reward for low confidence: 1,550 ± 161 JPY, for high confidence: 1,470 ± 258 JPY; paired $t$-test statistics: $t_9 = 0.797$, $P = 0.45$).

Considering that we simultaneously used MVPA patterns in as many as four frontoparietal ROIs in the neurofeedback sessions, the question arises as to whether some of them were more critical than others in changing confidence behaviourally. We explored the contribution of frontal versus parietal areas based on participants' relative success in the neurofeedback training in the different ROIs used (Fig. 9b). We found that for seven participants, DecNef induction success was highest in the parietal ROI, whereas for three participants, it was highest in the frontal ROIs. Interestingly, the effect size—mean net confidence changes—in each group were also very similar (paired $t$-test $t_8 = -0.433$, $P = 0.676$, IPL one-sample $t$-test against no change, $t_6 = 3.46$, $P = 0.0134$, LPFC one-sample $t$-test against no change, $t_2 = 2.34$, $P = 0.145$). Although these results are exploratory as they are limited by a small number of participants, they suggest that both parietal and prefrontal regions are of similar importance for the induced confidence changes in this study. We also analysed frontal ROIs separately (Supplementary Fig. 10b,c).

Finally, we addressed the question of whether DecNef induction may have led to behavioural changes via activity in brain areas other than the targeted regions. We conducted an 'information communication criterion' analysis (see the 'Methods' section and a previous study[23]). Conceptually, the analysis concerns how induction success in a particular ROI within the frontoparietal network may be predicted by, or associated with, multivoxel patterns in other ROIs. For instance, when participants activate a pattern of brain activity reflecting high confidence in a prefrontal region, in principle it is possible that it is accompanied by a pattern in V1/V2 also reflecting high confidence, and one could argue that the latter may be the ultimate cause of the behavioural change. The 'coefficient of determination' measure reported in Fig. 10 indicates the strength of such possible associations with a scale representing predictability. For all ROIs, as expected, the coefficient of the source ROI itself was always high (>70%, IPL Fig. 10a, inferior frontal sulcus (IFS; Fig. 10b), middle frontal sulcus (MFS; Fig. 10c), middle frontal gyrus (MFG; Fig. 10d)) and acted as a control. Importantly, the coefficient was low for ROIs that were not used in the neurofeedback paradigm (visual areas), as compared with the coefficients associated with ROIs that were concurrently used for induction. The same analysis was repeated with the averaged neurofeedback signal, that is, the overall likelihood fed back to participants on a trial-by-trial basis (Supplementary Fig. 11a). These results indicate that induction success was largely local for each ROI, and specifically undermine the possibility that DecNef changed confidence by influencing earlier visual areas.

Moreover, as univariate activations in the target frontoparietal regions where not found when participants successfully induced the target activation patterns, the latter alone caused confidence changes (Supplementary Fig. 11b–e).

## Discussion

In summary, using MVPA, correlates of confidence were found in frontoparietal areas; the patterns of classification differed for confidence, task accuracy and perceptual responses. Importantly, the mere induction of activation patterns in the same frontoparietal regions, via bi-directional neurofeedback, resulted in the corresponding bi-directional confidence changes. The bi-directional nature of the confidence changes was

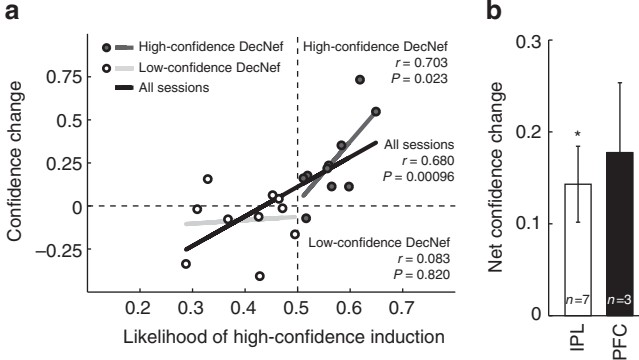

**Figure 9 | Correlation between neurofeedback success and confidence changes.** (**a**) Confidence rating changes between pre- and post-tests correlated with induction success on combined DecNef on day 2. The ordinate represents the change in confidence between pre- and post-tests. The abscissa, the likelihood of high-confidence induction, indexes neurofeedback success, 0.5 being the null point, with no effect expected. Values below 0.5 translate into higher likelihood of low-confidence induction. Each data point represents one participant— $n = 20$ data points, as all participants performed in two DecNef blocks—averaged across trials and runs performed on day 2 of each block. Correlations were inferred by computing Pearson's $r$. (**b**) Relative contributions to induction success. For each participant, in both high-confidence DecNef and low-confidence DecNef combined, the ROI with the overall highest induction success was selected. Bars represent the averaged net confidence change (absolute change in confidence) in participants for which the parietal ROI (IPL, $n = 7$) and one of the frontal ROIs (LPFC, $n = 3$) were selected. *$P < 0.05$, $P$ values corrected for multiple comparisons (Holm–Bonferroni). Centre values correspond to means, and error bars to s.e.m. ROI labels: IPL, inferior parietal lobule; LPFC, lateral prefrontal cortex.

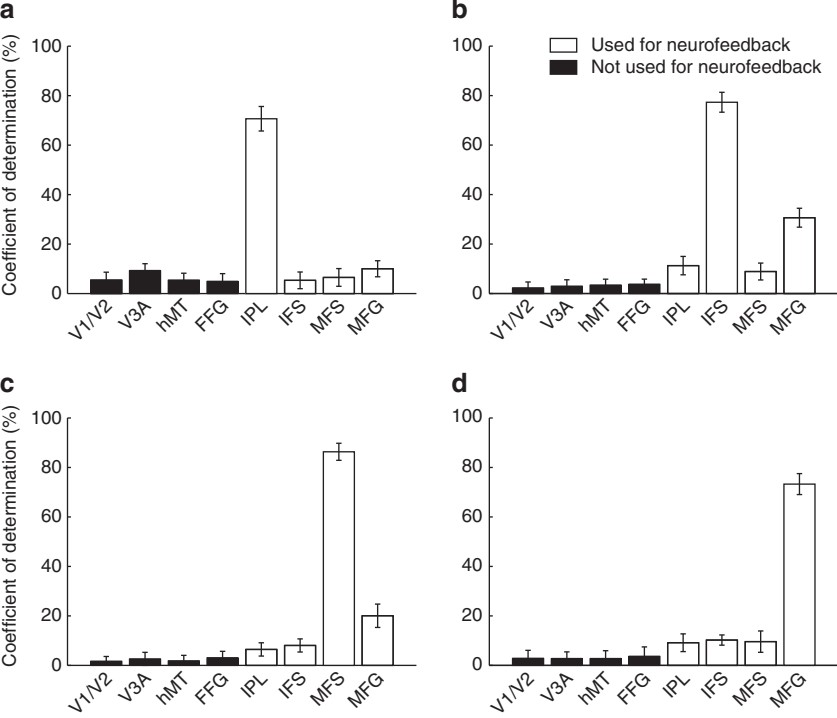

**Figure 10 | Information communication criterion analysis.** The information communication criterion analysis shows that induction of high- and low-confidence activation patterns in frontoparietal areas did not result from activation patterns in visual areas. The mean ( ± s.e.m.) coefficient of determination—goodness of fits between the likelihood in (**a**) IPL, (**b**) IFS, (**c**) MFS, (**d**) MFG and the predicted value for each area—multiplied by 100 for the sparse linear regression prediction by each of the activation patterns in V1/V2, V3A, hMT, FFG, IPL, IFS, MFS and MFG during DecNef, and from the IPL, IFS, MFS and MFG themselves as a control. The coefficient of determination is akin to variance-accounted-for (VAF). Centre values correspond to means, and error bars to s.e.m. FFG, fusiform gyrus; IFS, inferior frontal sulcus; IPL, inferior parietal lobule; MFG, middle frontal gyrus; MFS, middle frontal sulcus.

supported by three converging lines of evidence: (1) the binomial probability of observed versus expected changes at the individual level; (2) standardization of the second week effect by the ratio of week-2/week-1 effects; (3) the modelling analyses provided a formal framework, specifically through the sign and magnitude of the estimated parameters. It is notable that neurofeedback did not affect perceptual accuracy in the discrimination task, as this rules out the possibility that these representations of confidence may just reflect the strength of the internal perceptual signal. In addition, our psychophysical analyses indicate that the change in confidence due to DecNef was unlikely to be due to a change in response criterion.

Besides informing us about the nature of confidence, these results may also be relevant to the mechanisms of conscious perception[2,37]. Specifically, in the remarkable neurological condition of blindsight, patients with lesions to their primary visual cortex (V1) show residual visual capacity in their forced-choice responses in the absence of reported conscious awareness[38,39]. Importantly, in such cases, the patients also claim to be 'just guessing' when they make correct perceptual decisions. Therefore, congruent with our results, the existence of such patients suggests that confidence may be dissociable from perceptual decisions, and that confidence and awareness may be conceptually linked[2,40]. Compatible with this interpretation, several theories also suggest that the PFC (and to some extent also the parietal cortex) plays a central role in visual awareness[37,41,42].

Our MVPA results indicate that the correlates of confidence can be found in different frontal and parietal regions, and, importantly, that such correlates are multivariate patterns rather than overall activation levels. Furthermore, during

neurofeedback, when activation patterns were successfully induced, we did not find univariate activations in the target frontoparietal regions. This may seem incongruent with previous studies reporting univariate activations in prefrontal cortex associated with confidence[11,18,43]. One important difference is that in most of these previous studies, confidence responses had fixed associations with specific button presses. Thus, we cannot completely rule out the possibility that activations associated with high or low confidence may have simply reflected motor preparation in previous studies. In contrast, the MVPA in our study was derived from BOLD signal decoupled from the motor preparation, because the mapping changed from trial to trial and were only presented 4 s after stimulus offset.

In addition, contra the general view and a recent review claiming that perceptual content cannot be decoded in the PFC[44], here we have achieved classification of perceptual states from multivoxel activation patterns in PFC and parietal lobule. Indeed, electrophysiology data support the view that subregions of the PFC can represent both task-general and feature-specific perceptual content, even when the relevant stimuli or features were not attended to[45,46]. Although relatively rare, some fMRI studies that used multivariate decoding approaches have shown that visual information can be retrieved from frontoparietal areas[47,48]. The positive findings reported here are thus in line with these studies. In particular, several aspects in our methods may help explain why we succeeded in decoding perceptual content. First, we used an ROI-based approach (rather than a searchlight approach, which in some circumstances can be inefficient), and then utilized SLR as classification algorithm, a special case of logistic regression adapted to data sets where information is sparsely represented in the multidimensional feature space. To reduce the dimensionality of

the feature space, SLR is grounded on the assumption that only a small part of the feature is relevant to the classification problem being explored. On the other hand, support vector machines (SVMs) use kernels and therefore do not work directly on the features, but rather on a reduced and tractable representation. As a speculative possibility, perhaps SLR-based decoding is more suited for PFC where, unlike in visual areas (for example, V1; refs 49,50), there is probably no intrinsic columnar structure for the relevant representations. Moreover, PFC is implicated in a wide range of functions, as opposed to striate and extrastriate cortex, where all the processing is dedicated to visual stimuli. This non-uniformity in function, reflected by the distribution of relevant neural substrates within PFC, may well be an additional reason for the better performance of SLR. Further studies are needed to investigate these possibilities.

From a methodological viewpoint, our results are also of great interest because we have successfully induced a change in confidence after a comparatively short DecNef training session. Previously, in Shibata et al.[23] the neurofeedback training lasted 5 and 10 days. One possible interpretation is that in this previous study, induction of activation patterns was done in visual/perceptual processing areas, involving a single ROI, while in our study the targeted brain areas were multiple frontoparietal regions. In line with this view, in a recent real-time fMRI neurofeedback experiment, deBettencourt et al.[25] were able to use a single day of neurofeedback to train attention in informed participants, while their target regions of neurofeedback covered as much as the entire brain volume.

Overall, our results may seem to be at odds with the currently popular normative approach of studying confidence in psychophysics and electrophysiology, which defines a confidence rating as an optimal assessment of the strength of perception from a Bayesian perspective[19,51]. Our results suggest that confidence is dependent on late-stage processes, rather than a direct transformation of the perceptual signal[6,7,11], coded by the same neuronal substrate active in the perceptual decision-making process[7]. Congruent with this view, it has recently been reported that TMS to the premotor cortex also affected confidence reports, even though such activity likely took place downstream from perceptual decision-making processes[21,22]. Other studies are also congruent with the view that confidence depends on late-stage processes that are likely located in prefrontal regions[9,20,52,53].

Could one still argue that neurofeedback only changed the 'readout' of confidence, rather than confidence per se? We feel that this possibility is probably difficult to address empirically. For confidence to be a useful concept in experiments, it would be advantageous to be able to measure it behaviourally. In a nearly unfalsifiable manner, one could always challenge that any manipulation only concerns the measurement or reports, but not the phenomenon itself. However, three lines of thought may argue against this interpretation in the present case. First, we note that neurofeedback successfully shifted confidence bi-directionally, in the expected directions, even though confidence reports were uncoupled from fixed motor responses due to our task design. Furthermore, the analysis of meta-d′ suggests that this change cannot be explained simply in terms of a shift in criterion of responding strategy. Finally, the participants were not aware of the specific content of the representations induced by neurofeedback; as in previous DecNef studies[23,24,26] they were unable to tell which experimental group they were in even when asked in a forced-choice manner. These findings together suggest that neurofeedback changed the confidence representation itself, as well as possibly the reporting mechanism, albeit to a limited extent. Future studies could address this issue by, for instance, testing whether such change will result in different ways participants may perform cue integration. It is known that

rational observers would weight sensory cues based on their reliability[54,55]; if neurofeedback changed the subjectively assessed reliability (that is, confidence) of dot-motion-related decisions, they may be weighted down accordingly.

The mechanism regulating the DecNef-induced selective changes in confidence in specific directions remains unclear. Our interpretation is that DecNef is driven by association through stimulus contingency and neural operant conditioning[56]. Since we presented noise dot motion for 2 s either during the induction or during the following resting periods of neurofeedback, and participants received a reward based on their induction performance, over time, dot-motion stimuli likely became paired with the targeted brain activity pattern representing a specified level of confidence[26]. During post-DecNef behavioural tests, we reasoned that when dot motion was presented, the corresponding neural pattern would be re-triggered. However, we cannot rule out the possibility that DecNef-induced neural patterns might have carried over to the post-DecNef psychophysical test period as a form of spontaneous brain activity[57,58], without a single, particular association with dot motion. Future experiments can test for this possibility by having multiple stimulus conditions, for example, colour, dot motion, Gabor orientation, with only one stimulus set being presented during DecNef training to test for the specificity of its effects.

In conclusion, this study gives strong support to the view that confidence emerges as a late-stage metacognitive process. We demonstrated with a novel and powerful neurofeedback technique that we can manipulate perceptual confidence bi-directionally without changing task performance. And, importantly, this was achieved without the participants' knowledge of the purpose of the manipulation. This adds to the growing body of evidence on how confidence is generated in the brain. Although several issues remain to be addressed by future studies, this represents a promising step towards a new approach for elucidating the nature of metacognition.

## Methods

**Overall experimental design.** The entire experiment consisted of six neuroimaging sessions: retinotopy, a multivoxel pattern analysis (MVPA) session, DecNef block 1 (two sessions, that is, two consecutive days), and DecNef block 2 (two sessions, that is, two consecutive days; Fig. 2a). Each participant underwent DecNef training twice, once for high-confidence neurofeedback and once for low-confidence neurofeedback. DecNef blocks were separated by at least 1 week, and the order (high confidence first versus low confidence first) was counterbalanced across the participants. The experimenter was not blinded to group allocation. Before and after each DecNef block, the participants performed a psychophysical test outside the scanner. Their behavioural performance in these tests is our primary dependent variable of interest.

**Participants.** Eighteen participants (23.7 ± 2.5 years old; four females) with normal or corrected-to-normal vision participated in the first part of the study (retinotopy mapping and MVPA). One participant had to be removed owing to corrupted data. Participants were screened out either if they could not or declined to come back for the DecNef sessions, or due to low decoding accuracy for confidence in frontoparietal ROIs (less than 55% in more than two ROIs). Ten participants (24.2 ± 3.2 years old, three females) thus attended all ensuing neurofeedback training experiments. The results presented in the MVPA part are from the 17 participants that attended the initial MVPA session, while DecNef results are from the 10 participants that completed the whole experimental timeline. The number of participants to complete all sessions was predetermined based on our pilot study.

All of the experiments and data analyses were conducted at the Advanced Telecommunications Research Institute International (ATR). The study was approved by the Institutional Review Board of ATR. All the participants gave written informed consent.

**Apparatus and stimuli.** Visual stimuli were presented on an LCD display (1,024 × 768 resolution, 60 Hz refresh rate) during titration and the pre- and post-test stages, and via an LCD projector (800 × 600 resolution, 60 Hz refresh rate) during fMRI measurements in a dim room. All the stimuli were created and presented with Matlab (Mathworks) using the Psychophysics Toolbox extensions Psychtoolbox 3 (ref. 59). Stimuli were shown on a black background and consisted

of random dot motion (RDM). We used the Movshon–Newsome (MN) RDM algorithm[60], in which three uncorrelated random dot sequences are generated and frames from each are interleaved to form the presented motion stimulus. For each set, the probability that a dot is replotted in motion—as opposed to randomly replaced—is given by the coherence value. Dots replotted in motion are defined as 'signal' dots. This routine generates a set of dots as a $N_{dots} \times 2$ matrix of locations, and then plots them. In plotting the next set of dots (for example, set 2), it prepends the preceding set (for example, set 1). The RDM stimulus was created in a square region of $20 \times 20$ deg, but only the region within a circular annulus was visible (outer radius: 10 deg, inner radius: 0.85 deg). Dot density was $0.5\,\mathrm{deg}^{-2}$ (contrast 100%), with a speed of $9\,\mathrm{deg\,s}^{-1}$ and size of 0.12 deg. Signal dots all moved in the same direction (left or right, non-cardinal directions of 20 deg and 200 deg), whereas noise dots were randomly replotted. Dots leaving the square region were replaced with a dot along one of the edges opposite to the direction of motion, and dots leaving the annulus were faded out to minimize the edge effects.

**Behavioural task for fMRI scans for MVPA and pre-/post-tests.** Participants performed the same behavioural task for the fMRI scans for MVPA and pre/post-tests in the neurofeedback sessions. The task was a two-choice perceptual discrimination task with a confidence rating using RDM stimuli. A pair of pre- and post-tests was conducted for each DecNef session (high/low confidence). The pre-tests were conducted before the neurofeedback session on day 1, and post-test tests after the neurofeedback session on day 2 (see Fig. 2a).

Each trial started with a 1 s fixation period. A noise RDM (0% coherence) was then presented for 1 s, followed by the stimulus for 2 s. Critically, the transition from noise to stimuli was smooth and indistinguishable. After a delay of 4 s, the participants were given 2 s to report the direction of motion (left or right) by pressing one of two keys on a keyboard (psychophysical testing) or a response pad (fMRI scans), and two additional seconds to report their confidence about their decision by pressing one of four keys (four-point scale, 1 corresponding to 'guess', 4 to 'totally sure'). At the end of each trial, an intertrial-interval (ITI) was inserted (2 s for pre- and post-tests, 6 s for fMRI scans), consisting of a black background.

For fMRI scans, the participants completed 192 trials (16 trials per run, in 12 runs). In the pre/post-tests, the participants completed up to 216 trials subdivided into three runs on each day, the mean ($\pm$ s.e.m.) was $195 \pm 6$ trials completed on average. To minimize motor-response confounds in fMRI signals, in both settings the response keys corresponding to each possible choice in both the perceptual and confidence reports were indicated on the screen and their positions were randomized across trials.

The task difficulty was adjusted for each participant by titrating the coherence of RDM before the fMRI scans for MVPA. After 16 practice trials, the coherence of the RDM was individually titrated to estimate perceptual threshold using an adaptive staircase method (QUEST)[61], outside the scanner. Trials from two parallel QUEST tracks (40 trials each) were randomly interleaved. The coherence was estimated to yield accuracy of 70 and 80% in each track. During the fMRI scans for MVPA, to decode confidence level while maintaining consistent performance, the coherence level was adjusted at the end of each fMRI run in the event that the participant's performance varied more than 10% from a mean of 75% correct, to ensure that task difficulty was kept constant throughout the experiment. Mean ($\pm$ s.e.m.) coherence levels during the fMRI MVPA session were $13.3\% \pm 2.9\%$ and $8.0\% \pm 2.5\%$ for the upper and lower bounds, respectively. In the beginning of the first pre-test, the coherence level was re-determined with QUEST. The re-determined coherence level was then used during the rest of the pre- and post-tests.

In each experimental session (fMRI scans for MVPA and pre/post tests), 62.5% of all the trials had a coherence level around perceptual threshold of 75% accuracy, 12.5% of trials had a very high coherence (80%) and the remaining 25% trials had noise of 0% coherence. Specifically, for the trials at perceptual threshold, the coherence level in each trial was randomly drawn from a linear sequence within the interval of coherence levels corresponding to 70 and 80% correct, hence expected to yield accuracy of 75% on average. In each trial, one of two motion directions (20 or 200 deg; Fig. 2b) was presented at one of three different possible coherence levels (0%, threshold, or high). The order of presentation of the orientations and coherence levels was randomized across trials. Throughout the task, the participants were asked to fixate on a white cross (size 0.5 deg) at the centre of display.

**fMRI scans for retinotopy.** On the first day, we measured participants' retinotopic maps to functionally define visual cortical areas and motion sensitive areas individually, using standard retinotopic methods with blood-oxygen-level-dependent (BOLD) signal[62]. Travelling-waves methods involve the sequential presentation of stimuli that induce travelling waves of activity in the primary visual cortex[62]. Three types of stimuli were used: a set of rings of increasing radius to measure the eccentricity maps; set of wedges, with the tip at the centre of gaze but extending in different directions to measure angle maps; a set of moving dots, where the direction of motion was randomly switching from in and out of the centre of gaze, to define motion area hMT. In addition, the participants were presented with a reference stimulus to localize the retinotopic regions in V1/V2 corresponding to the visual field stimulated by the RDM. The reference stimulus was composed of a

coloured checkerboard pattern presented within an annulus subtending 0.85 to 20 deg from the centre of a grey screen.

**fMRI scans for MVPA.** The purpose of the fMRI scans in the MVPA session was to obtain the fMRI signals corresponding to different behavioural measures (for example, high- and low-confidence states). These behavioural measures would then be used as labels to compute the parameters for the decoders used in the MVPA and DecNef blocks[23]. During the MVPA session, the participants performed a perceptual two-choice discrimination task with a confidence rating in the fMRI scanner. The participants discriminated the direction of RDM with various coherence levels, then rated their confidence (see subsection 'Behavioural task for fMRI scans for MVPA and pre-/post-tests'). We thus obtained BOLD signal patterns (see subsection 'fMRI scans preprocessing' given below) for all of the behavioural measures associated with the task at various levels of coherence. Throughout the fMRI runs, the participants were asked to fixate on a white cross— size 0.5 deg—presented at the centre of the display. A brief break period was provided after each run on the participant's request. Each fMRI run consisted of 16 task trials (1 trial = 18 s; Fig. 2b), with a 20 s fixation period before the first trial (1 run = 308 s). The entire session consisted of 12 runs. The fMRI data for the initial 20 s of each run were discarded due to possible unsaturated T1 effects. During the response period, the participants were instructed to use their dominant hand to press the button on a diamond-shaped response pad. Concordance between responses and buttons was indicated on the screen and, importantly, randomly changed across trials to avoid motor preparation confounds (that is, associating a given response with a specific button press).

**fMRI scans preprocessing.** The fMRI signals in native space were preprocessed using custom software (mrVista software package for MATLAB, freely available at http://vistalab.stanford.edu/software/). The mrVista package uses functions and algorithms from the SPM suite (freely available at http://www.fil.ion.ucl.ac.uk/spm/). All the functional images underwent three-dimensional motion correction. For retinotopic scans, we applied slice-timing correction, averaged runs across stimuli groups and computed a coherence analysis. No spatial or temporal smoothing was applied. The rigid-body transformations were performed to align the functional images to the structural image for each subject. A grey-matter mask was used to extract fMRI data only from grey-matter voxels for further analyses. The boundaries between the retinotopic areas V1, V2, V3A and the sub-region that corresponded to the reference stimulus within V1/V2, were identified using the standard visual field mapping procedure[62], and a motion localizer was used to functionally define hMT. Other regions of interest (ROIs), such as the fusiform gyrus and parts of the lateral prefrontal cortex (lateral PFC) and parietal areas, were anatomically defined through cortical reconstruction and volumetric segmentation using the Freesurfer image analysis suite, which is documented and freely available for download online (http://surfer.nmr.mgh.harvard.edu/). More specifically, the prefrontal ROIs of the lateral PFC were the IFS, the MFS and MFG. Once the ROIs were identified, time courses of BOLD signal intensities were extracted from each voxel in each ROI and shifted by 6 s to account for the hemodynamic delay using the Matlab software. A linear trend was removed from the time course, and the time course was $z$-score normalized for each voxel in each run to minimize the baseline differences across runs. The data samples for computing the MVPA were created by averaging the BOLD signal intensities of each voxel for three volumes, corresponding to the 6 s from stimulus onset to response onset.

**Algorithm for MVP analyses.** We used sparse logistic regression (SLR), which automatically selects the relevant voxels in the ROIs for MVPA[63], to construct multiple binary classifiers based on the three main behavioural variables of interest: confidence (high versus low), discrimination accuracy (correct versus incorrect) and motion perception (left versus right). We used SLR, as opposed to support vector machines (SVMs) or other well-known classifying algorithms, for two main reasons pertaining to the design of the study. First, when the number of features is much greater than the number of samples, SLR is known to be advantageous[63]. Furthermore, and more importantly, SLR concomitantly outputs a categorical variable (the class), as well as its probability. For the neurofeedback stage, this aspect is significant, as the signal fed back to the participants can therefore directly be the probability that the current activation patterns belong to a given class.

The decoder, as a statistical classifier, uses the LDF to separate two classes, $S_1$ and $S_2$. The LDF is represented by the weighted sum of each feature value (voxels),

$$f(x, \theta) = \sum_{d=1}^{D} \theta_d x_d + \theta_0 \qquad (1)$$

where $x = (x_1, ..., x_D)^t \in \Re^D$ is the input feature vector (the voxels from the fMRI scans) in $D$ dimensional space, and $\theta = (\theta_0, \theta_1, ..., \theta_D)^t$ is the weight vector (including a bias term, $\theta_0$). Therefore, the hyperplane where $f(x, \theta) = 0$ represents the boundary between the two classes. Logistic regression allows us to calculate the probability that an input feature (a given sample) belongs to category $S_2$ through the logistic function,

$$p = \frac{1}{1 + \exp(-f(x, \theta))} \equiv P(S_2 | x) \qquad (2)$$

Note that $p$ ranges from 0 to 1, and is equal to 0.5 when $f(x, \theta) = 0$ (on the boundary) and approaches 0 or 1 when $f(x, \theta)$ tends towards plus or minus infinity (far away from the boundary). Since the number of samples is fewer than the number of features (voxels), logistic regression is not directly applicable to the data set. Therefore, a dimensionality reduction was implemented by pruning out irrelevant voxels through automatic relevance determination, thereby treating the whole data matrix as sparse. For the original and more detailed version, please refer to Yamashita *et al.*[63]

**MVP analyses data sets and cross-validation.** For unbalanced data sets (that is, task accuracy—correct versus incorrect trial classification), we used a weighted samples method for the majority group in the training data sets. That is, by design the proportion of correct to incorrect trials is ~3:1, and therefore each sample in the correct trials group is automatically assigned a weight <1 (~0.33), to penalize the sample size bias for the differential ratios.

For each MVPA, we performed a $k$-fold cross-validation, where the entire data set is repeatedly subdivided into a 'training set' and a 'test set'. The two can be seen as independent data sets that are used to fit the parameters of a model (decoder) and evaluate the predictive power of the trained (fitted) model, respectively. For each behavioural variable of interest, and for each participant the number of folds was automatically adjusted between $k = 9$ and $k = 11$ to be a (close) divisor of the number of samples in the data set. Thus, the number of folds in each cross-validation procedure was ~10, a typical value for $k$-fold cross-validation procedures[64]. The $k$-fold cross-validation procedure is critical in any predictive classification problem as it serves the purpose of limiting overfitting, and gives an index of the generalizability of the model[64,65]. Furthermore, the SLR-based classification was optimized by using an iterative approach. That is, during each fold of the cross-validation, the process was repeated $n$ times ($n = 10$). On each iteration, the selected features were removed from the pattern vectors, and only features with unassigned weights were used for the next iteration. At the end of the $k$-fold cross-validation, the test accuracy was averaged for each iteration across folds, to evaluate the accuracy at each iteration. The optimal number of SLRs (number of iterations) was thus chosen and used for the final computation of the decoder used in the neurofeedback training procedure. Importantly, the confidence decoder used in the DecNef was constructed by using only correct trials, as compared with the main analysis for which both correct and incorrect trials were used. Furthermore, the calculation of weights used for DecNef was done using the entire data set and the optimal number of iterations (as described above), while we did cross-validation (using a subset of the data for weights' calculation) to measure decoding accuracy.

**Confidence data set.** Confidence was rated on a four-point scale, and hence we reassigned the intermediate levels (2, 3) to both the low- and high-confidence classes to collapse the four initial confidence levels to two levels, and equate the number of trials in each class. For each participant we first merged one intermediate class with the high- or low-confidence class depending on the total number of trials. Then, to equate the number of trials, we added randomly sampled trials from the left-out intermediate confidence level to the confidence class now having a lower total number of trials. This re-balancing was based on the confidence rating response distribution, and on the final number of trials, and was repeated $n$ times ($n = 10$, owing to the low number of resampled trials). Given these sampling sets the main analysis reported in the 'Results' section was performed as follows. To directly compare the information contained in multivoxel patterns pertaining to the confidence dimension across various ROIs, the best sample set was voted by $k$-fold cross-validation mean accuracy, specifically optimizing for frontoparietal ROIs, given the a priori assumption that these areas are critical for generating confidence. Therefore, once a sample set was selected, the cross-validation mean accuracy for that particular set was used for each ROI, thus ensuring that the comparison would pertain to exactly the same samples and the information they contained. The rationale behind this approach is that if a multivoxel pattern is found for discriminating confidence in frontoparietal areas, does this same sample set also contain confidence-related information in visual processing areas? We also ran a more conservative analysis, where all the sample sets were averaged after taking the mean cross-validated accuracy for each set, and the same pattern of results was found; namely, decoding of confidence was higher in frontoparietal areas as compared with visual processing regions (Supplementary Fig. 1).

**Rectification analysis.** In addition, in the rectification analysis, we proceeded as follows. To predict two classes $S_1$ and $S_2$, the decoder evaluates a certain linearly weighted voxel value, the LDF. The LDF is a continuous variable, and for each data sample the decoder then binarizes it into a categorical variable. When the value of the continuous variable is positive, the decoder predicts class $S_2$, and when it is negative, the decoder predicts class $S_1$, the boundary being at zero. If the relationship between two binary measures (in this case, from perceptual responses to accuracy, or confidence) is nonlinear, a decoder with the LDF built from the first will not predict the second. Thus, in the rectification analysis, we apply a nonlinear transformation such that $\text{LDF}_{new} = \text{abs(LDF)}$ before using the new LDF value to assign a label to a given sample (trial). The new boundary for discriminating the

two classes is assessed for each ROI, with an iterative approach ($i = 200$, step $= 0.1$, start threshold $= 0.1$). In the correct versus incorrect rectification, because of the imbalance in the two categories, we performed a sampling procedure to randomly create equally sized sample sets. The procedure was repeated $n$ times ($n = 50$). The iterative boundary evaluation process between the two classes described above was repeated for each fold (sample set). Finally, an optimal set (with equal sample sizes) is obtained with an optimal boundary between the two classes LDF distributions, for each ROI and participant. In addition, we also performed the analyses where the optimal boundary is evaluated at the group level, as well as taking the average of all the sampling folds and thresholds. These methods being not equally conservative, we report the accuracies of the rectifications averaged between the three methods outputs, thus providing a good measure of the information contained in the multivoxel patterns pertaining to both dimensions (perceptual information and task accuracy). In the case of rectification to confidence, only the sample set used in the confidence decoder (the set selected for optimal decoding in frontoparietal areas) was used. This ensured that the exact same samples were used for all the ROIs. Importantly, for both rectification analyses, the weights from the motion perception decoder were obtained by using all the trials at threshold coherence, to ensure that extreme values of coherence were not the main drivers for the analysis. Specifically, that different coherence values, naturally associated with a larger fraction of correct or incorrect responses (high coherence with correct responses, and low coherence with incorrect responses), would not trivialize the analysis.

**ROIs and voxels for classification.** Each binary decoder was trained to classify a pattern of BOLD signals into one of the above mentioned classes using data samples obtained from up to 120 trials with threshold coherence (out of a maximum of 192 total trials), collected in up to 12 fMRI runs. As a result, the inputs to the decoders were the participants' moment-to-moment brain activations, while the outputs of the decoders represented the calculated likelihood of each behavioural measure. The mean ($\pm$ s.e.m.) number of voxels for decoding was $690 \pm 20$ for V1/V2, $569 \pm 45$ for V3A, $255 \pm 21$ for hMT, $517 \pm 16$ for FFG, $1,802 \pm 63$ for IPL, $490 \pm 19$ for IFS, $412 \pm 16$ for MFS and $1,350 \pm 42$ for MFG.

For the control analysis with equal number of voxels in each ROI, the data samples were created with the following steps. For each voxel, estimated BOLD signal amplitude and its $t$-value were computed on the basis of a univariate GLM in a contrast between stimulus presentation and a blank baseline (the ITI). Then, voxels were sorted according to their amplitudes ($t$-values). For each participant, the ROI with the lowest number of voxels was determined (lowest number of voxels $= N_{min}$). For all the other ROIs, we then selected the $N_{min}$ most significant voxels to run the control MVPA.

**Neurofeedback sessions.** Each DecNef block consisted of two consecutive days of fMRI scanning, during which the participants implicitly learned to induce brain activation patterns corresponding to high or low confidence. Each participant did both high- and low-confidence DecNef, and the order of confidence inductions was counterbalanced across the participants (that is, high then low versus low then high). After each scanning session, the participants were asked to describe their strategies in making the disc size larger. The answers varied from 'I was counting', to 'I was focusing on the disc itself', to 'I was thinking about food' (see Supplementary Table 6). At the end of the experiments, the participants were asked to which group they thought they were assigned to. The participants could answer only at chance ($n = 5$, 2 months later; and $n = 4$, 5 months later—one participant could not be joined; Chi-square test, $\chi^2 = 0.225$, $P = 0.64$).

On each day of a given DecNef block, the participants engaged in up to 11 fMRI runs. The mean ($\pm$ s.e.m.) number of runs per day was $10 \pm 0.1$ across sessions and participants. Each fMRI run consisted of 16 trials (1 trial $= 20$ s) preceded by a 30 s fixation period (1 run $= 350$ s). The fMRI data for the initial 10 s were discarded to avoid unsaturated T1 effects. Throughout a run, the participants were instructed to fixate their eyes on a white bull's-eye at the centre of a white disc (0.75 deg radius) presented at the centre of the display. After each run, a brief break period was provided on the participant's request. Each trial (Fig. 6) consisted of an induction period (6 s), a fixation period (6 s), a feedback period (up to 2 s) and an ITI (6 s), in this order.

The participants were instructed to regulate their brain activity during the induction period, with the goal of making the size of a solid white disc, presented later in the feedback period, as large as possible. The experimenters provided no further instructions or strategies. During the fixation period, the participants were asked to simply fixate on the central point and rest. This period was inserted between the induction and the feedback periods to account for the hemodynamic delay, assumed to last 6 s. Either during the induction period, or at the beginning of the fixation period (pseudo-random onsets: 2, 4, 6 or 8 s from trial start) a 2 s noise RDM was also presented. Pseudo-random onsets were selected to ensure minimal interference and maximal effect of the RDM on the induction process. Subsequently, during the feedback period, a grey disc was presented for up to 2 s. The size of the disc displayed in the feedback period represented how much the current BOLD signal patterns obtained in the induction period corresponded to activation patterns measured when the participants were in a given confidence state, while performing the perceptual task in the MVPA session. The grey disc was always enclosed in a larger white concentric circle (5 deg radius), which indicated the disc's maximum possible size. The feedback period was followed by an ITI that

lasted 6 s, during which the participants were asked to fixate on a central white point and rest. This period was followed by the start of the next trial.

The size of the disc presented during the feedback period was computed at the end of the fixation period with the following steps[23]. First, measured functional images during the induction period underwent three-dimensional motion correction using Turbo BrainVoyager (Brain Innovation). Second, time-courses of BOLD signal intensities were extracted from each of the voxels identified in the MVPA session for each of the four ROIs (IPL, IFS, MFS and MFG), and were shifted by 6 s to account for the hemodynamic delay. Third, a linear trend was removed from the time course, and the BOLD signal time course was $z$-score normalized for each voxel using BOLD signal intensities measured for 20 s starting from 10 s after the onset of each fMRI run. Fourth, the data sample to calculate the size of the disc was created by averaging the BOLD signal intensities of each voxel for 6 s in the induction period. Finally, the likelihood of each confidence state was calculated from the data sample using the confidence decoder computed in the MVPA session. The size of the disc was proportional to the averaged likelihood (ranging from 0 to 100%) of the target confidence level (high/low) assigned to each participant on a given DecNef block from the four different frontoparietal ROIs. Importantly, the participants were unaware of the relationship between their activation patterns induction and the size of the disk itself. The target confidence was the same throughout the DecNef block. In addition to a fixed compensation for participation in the experiment, a bonus of up to 3,000 JPY was paid to the participants based on the mean size of the disc on each day.

**Offline analyses.** The likelihood of inducing the targeted confidence level was taken as the measure of neurofeedback success (since it is a binary problem, we can denote the likelihood of high-confidence induction as $p$, and it follows that the likelihood for low-confidence induction can be labelled as $1 - p$). Furthermore, the induction success was estimated with the 'best' ROI; that is, the ROI among the four used for neurofeedback that showed the highest contribution to successfully inducing the target confidence state. The data were averaged for each participant across all the trials and all runs from each day. In the main text, we report results from the session immediately preceding the post-test for each DecNef training group (second day). The same analysis applied to day 1 of the neurofeedback training for aggregate DecNef conditions (high- and low-confidence DecNef, see Supplementary Fig. 2) correlated with confidence changes between pre- and post-test ($n = 20$, Pearson's $r = 0.701$, $P = 0.0012$), but not when taken singularly (high-confidence DecNef, $n = 10$, Pearson's $r = 0.446$, $P = 0.229$; low-confidence DecNef, $n = 10$, Pearson's $r = 0.010$, $P = 0.979$). A control analysis, carried out by using the averaged likelihood of the four frontoparietal ROIs yielded comparatively similar results. As in the main analysis reported in the 'Results' section, induction success on the second day of the neurofeedback training for each DecNef condition (high and low confidence), that is, the session immediately preceding post-test, correlated with the change in confidence ($n = 20$, Pearson's $r = 0.562$, $P = 0.0099$). Similarly, this effect was mainly driven by the high-confidence DecNef condition ($n = 10$, Pearson's $r = 0.870$, $P = 0.0012$); the correlation was not statistically significant for low-confidence DecNef alone ($n = 10$, Pearson's $r = 0.061$, $P = 0.867$).

To investigate the relative contributions of the frontal and parietal brain regions based on the 'best' ROI, a further analysis was conducted. First, for each participant each ROI was assigned a value given by the mean of the daily averaged induction success across 4 days of neurofeedback (high-confidence and low-confidence DecNef, each comprising 2 days of neurofeedback). Then, ROIs from the frontal region were pooled together, and likewise for the parietal region, giving a single value representing each region for each participant. Finally, the participants were assigned to either the 'parietal' group or the 'frontal' group, based on the maximum of the computed values.

We performed one further offline test, the 'information communication criterion' analysis: using a sparse linear regression to predict a neurofeedback signal in each of the frontoparietal ROIs (the likelihood of the target confidence during the induction stage), from an activation pattern in each of the seven other areas (four visual areas, and the three remaining frontoparietal ROIs)[23]. The activation pattern from the ROI itself acted as control. A predicted value was obtained as the linearly weighted sum of the voxel activities in each area. Prediction accuracy was defined as a coefficient of determination and evaluated by a leave-one-day-out cross-validation procedure. That is, data measured on one day during the induction stage were treated as the test data while data measured on the remaining days were used for training the sparse linear regression decoder to predict trial-by-trial likelihoods in the target ROI (one each of IPL, IFS, MFS or MFG). Four cross-validation sets were thus generated for each ROI. The coefficient of determination here indicates the proportion of variability in the likelihoods on a trial-by-trial basis in the target ROI that is explained by voxel activities in each other area. The coefficient of determination for each area was first averaged over the cross-validation sets and then across the participants. The analysis was repeated for the neurofeedback signal given by likelihood averaged across the four frontoparietal ROIs (representing the actual feedback to the participants).

Last, we ran a control GLM analysis contrasting successful versus unsuccessful induction trials during neurofeedback blocks to investigate whether overall increase or decrease in activation in the target regions could explain the obtained results. For this analysis, a GLM model was fitted to each voxel, separately for each day

(session) and DecNef block (high- and low-confidence DecNef). The trials considered as successful had an induction likelihood $P > 0.5$, unsuccessful trials had a likelihood $P \leq 0.5$, Betas were extracted from each voxel and averaged across target ROIs (IPL, IFS, MFS, MFG). Reported results are the mean ($\pm$ s.e.m.) betas across the participants.

**Nonlinear mathematical modelling.** We constructed a mathematical model to objectively examine the effects of high- and low confidence DecNef, a persistence of learning (of DecNef effect) owing to 1-week elapse, and the weaker second week effect. The model was fit to four measurement values of perceptual confidence at the four time points for each participant and possesses four model parameters. The $X_j^i$ variables represent the experimentally measured confidence values, while $\hat{X}_j^i$ are the estimated confidence values, for each participant (with $i = 1{:}10$; that is, behavioural outcome of DecNef effects) at each time point (with $j = 1{:}4$); The main four-parameter nonlinear model to describe DecNef effects is formally outlined as follows.

$$\hat{X}_1^i = B \tag{3}$$

for $i = 1{:}5$, group low–high

$$\hat{X}_2^i = B + \varepsilon \cdot \Delta \tag{4}$$

$$\hat{X}_3^i = B + \varepsilon \cdot \Delta \cdot \alpha \tag{5}$$

$$\hat{X}_4^i = B + \varepsilon \cdot \Delta \cdot \alpha + \gamma \cdot \Delta \tag{6}$$

for $i = 6{:}10$, group high–low

$$\hat{X}_2^i = B + \Delta \tag{7}$$

$$\hat{X}_3^i = B + \Delta \cdot \alpha \tag{8}$$

$$\hat{X}_4^i = B + \Delta \cdot \alpha + \varepsilon \cdot \gamma \cdot \Delta \tag{9}$$

$$\text{Error} = \Sigma^i \Sigma^j \left( X_j^i - \hat{X}_j^i \right)^2 = F(\alpha, \varepsilon, \gamma, \Delta) \tag{10}$$

where $B$ is the initial baseline (0—the realigned confidence level); $\Delta$, the change in confidence by high-confidence DecNef in the first week; $\varepsilon$, the ratio of low-confidence DecNef effect normalized by high-confidence DecNef effect; $\gamma$, the reduced second-week DecNef effect; and $\alpha$, the learning persistence during the week-long interval. Submodels are defined by setting different parameters to 0 or 1, one at a time or concomitantly following complexity logic. This gives rise to a hierarchical group of models, from simpler to more complex (capturing single or increasingly more aspects of DecNef effects on confidence). The first model is the simplest and only estimates $\Delta$, with the other parameters setting as $\varepsilon = 0$, $\gamma = 1$, $\alpha = 1$. The second model, by complexity order, assumes up and down-DecNef effects, estimates $\Delta$ and $\varepsilon$, while $\gamma = 1$, $\alpha = 1$. The third model estimates $\Delta$, $\varepsilon$ and $\gamma$, with $\alpha = 1$. Further DecNef-based models estimate $\Delta$, $\varepsilon$ and $\alpha$, with $\gamma = 1$; or estimate $\Delta$, $\varepsilon$ and $\alpha$, with $\gamma = 0$; or estimate $\Delta$, $\varepsilon$ and $\gamma$, with $\alpha = 0$; or finally, estimate $\Delta$, $\gamma$, and $\alpha$, with $\varepsilon = 0$.

We considered alternative models that do not take into account DecNef direction assumptions or a posteriori conceptions. These are a one-parameter constant confidence model (confidence does not change, is constant throughout the experiment), with two versions: $k = \underline{X}_1^i$, or $k = \text{mean}(\underline{X}_2^i, \underline{X}_3^i, \underline{X}_4^i)$. Other free models are a within-week constant confidence, and within-week constant confidence with two or four additional linear parameters.

**Model comparison.** For model comparison, we used the AIC[33].

Raw AIC is computed according to the following equation:

$$\text{AIC} = n\log(\hat{\sigma}^2) + 2k \tag{11}$$

where $\hat{\sigma}^2 = \frac{\text{Residual Sum of Squares}}{n}$, $n$ is the sample size and $k$ the number of parameters in the model. In our set of global models, $n = 30$ (because the initial time point, B, was 0 and thus not considered), and $k$ varied from 1 to 4. In the modelling reported for small sample sizes (that is, $n/k < \sim 40$), the second-order or AICc should be used instead. Although the AICc formula assumes a fixed-effects linear model with normal errors and constant residual variances, while our models are nonlinear, the standard AICc formulation is recommended unless a more exact small-sample correction to AIC is known[34]:

$$\text{AICc} = \text{AIC} + \frac{2 \cdot k \cdot (k+1)}{(n-k-1)} \tag{12}$$

For comparing models, two useful metrics are $\Delta_{\text{AICc}}$ and Akaike weights ($w_i$). $\Delta^i_{\text{AICc}}$ is a measure of the distance of each model relative to the best model (the model with the most negative, or lowest, AIC value), and is calculated as:

$$\Delta^i_{\text{AICc}} = \text{AICc}_i - \min(\text{AICc}) \tag{13}$$

As indicated in Burnham and Anderson[34], $\Delta^i_{\text{AICc}} < 2$ suggests substantial evidence for the $i$-th model, while $\Delta^i_{\text{AICc}} > 10$ indicates that the model is implausible. Akaike weights ($w_i$) provide a second measure of the strength of evidence for each model,

are directly related to $\Delta^i_{\text{AICc}}$ and are computed as:

$$w_i = \frac{\exp\left(-\Delta^i_{\text{AICc}}/2\right)}{\sum_{i=1}^{R} \exp\left(-\Delta^i_{\text{AICc}}/2\right)} \tag{14}$$

AIC analysis results are reported in Supplementary Table 7, with values reported being AICc, $\Delta_{\text{AICc}}$ and Akaike weights ($w_i$).

In cases such as ours, where high degrees of model selection uncertainty exist (the best AIC model is not strongly weighted), a formal solution is to compute parameter estimates through model-averaging. For this approach, two procedures may be used, depending on the results. The first approach makes use of only a limited subset of models that are closest to the current best model ($\Delta_{\text{AICc}} < 2$), while the second approach will consider all models (in fact, this accounts to consider all models with $w_i \neq 0$). We applied the first approach, selecting only models with high likelihood. Parameters are estimated according to the equation:

$$\hat{\bar{\beta}} = \frac{\sum_{i=1}^{R} w_i \hat{\beta}_i}{\sum_{i=1}^{R} w_i} \tag{15}$$

where $\hat{\beta}_i$ is the estimate for the predictor in a given model $i$, and $w_i$ is the Akaike weight of that model.

Unconditional error, necessary to compute the unconditional confidence interval for a model-averaged estimate, can be calculated according to the following equation:

$$\widehat{se}\left(\hat{\bar{\beta}}\right) = \sum_{i=1}^{R} w_i \sqrt{\widehat{var}\left(\hat{\beta}_i\right) + \left(\hat{\beta}_i - \hat{\bar{\beta}}\right)^2} \tag{16}$$

Where $\widehat{var}\left(\hat{\beta}_i\right)$ is the variance of the parameter estimate in model $i$, and $\hat{\beta}_i$ and $\hat{\bar{\beta}}$ are as defined above. The confidence interval is then simply given by the end points:

$$\hat{\beta} \pm z_{1-\alpha/2}\widehat{se}\left(\hat{\beta}\right) \tag{17}$$

For a 90% confidence interval, $z_{1-\alpha/2} = 1.65$.

Note that the unconditional variance comprises two terms, the first one local (internal variance of model $i$), while the second one global, in that it represents the variance between the common estimated parameter and the true value in model $i$. Since these models are very stable and robust, changing the initial condition set does not lead to different solutions, thus providing proof that solutions reached are globally best. Therefore, to assess the local variance, we recreated surrogate data sets by selecting $k$ samples out of a population of $n$ samples, with all possible combinations. This is equivalent to the binomial coefficient, thus creating $\frac{N!}{K!(N-K)!}$ subgroups. For the estimation, $k = 7$, as the first non-even value for $k > n/2$ (size of each group). For each model, the parameters were independently estimated, and we thus calculated the population variance.

**Statistical analyses and computational routines.** All statistical analyses were performed with IBM SPSS for Windows, version 21.0, and MATLAB Version 7.13.0.564 (R2011b) (MathWorks), both with built-in functions as well as with functions commonly available on the MathWorks online repository or custom written code.

For MVPA results, mean decoding accuracies were averaged across the participants. Statistical analyses were performed with two-tailed $t$-tests against a chance accuracy of 50%. For multiple comparisons, we used the Holm–Bonferroni procedure (see below), and we report corrected $P$ values.

The effects of DecNef on behavioural data were statistically assessed using repeated measures ANOVA tests as well as two-tailed, or single-tailed were warranted, $t$-tests were utilized for comparisons.

For multiple comparisons, we used the Holm–Bonferroni correction, where the $P$ values of interest are ranked from the smallest to the largest, and the significance level $\alpha$ is sequentially adjusted based on the formula $\frac{\alpha}{(n-i+1)}$ for the $i$-th smallest $P$ values. In the text, for enhanced clarity, we present the results as corrected $P$ values.

We used Matlab optimization routines to solve our systems of nonlinear equations with a nonlinear programming solver, under least squares minimization. The Matlab solver was *fmincon*, utilizing the following optimization options. A sequential quadratic problem (SQP) method was used, specifically, the 'SQP' algorithm. This algorithm is a medium scale method, which internally creates full matrices and uses dense linear algebra, thus allowing additional constraint types and better performance for the nonlinear problems outlined in the previous section. As compared with the default *fmincon* 'interior point' algorithm, the 'SQP' algorithm also has the advantage of taking every iterative step in the region constrained by bounds, which are not strict (a step can exist exactly on a boundary). Furthermore, the 'SQP' algorithm can attempt to take steps that fail, in which case it will take a smaller step in the next iteration, allowing greater flexibility. We set bounded constraints to allow only certain values in the parameter space to be taken by the estimates. As such, boundaries were set as: $\Delta \in [0\ \text{Inf}]$, $\varepsilon \in [-1\ 0]$, $\gamma \in [0\ 1]$, and $\alpha \in [0\ 1]$. The function tolerance was set at $10^{-20}$, the maximum number of iterations at $10^6$ and the maximum number of function evaluations at $10^5$.

**MRI parameters.** The participants were scanned in a 3T MR scanner (Siemens, Trio) with a head coil in the ATR Brain Activation Imaging Center. The functional MR images for retinotopy, the MVPA session and DecNef sessions were acquired using gradient EPI sequences for measurement of BOLD signals. In all the fMRI experiments, 33 contiguous slices (TR = 2 s, TE = 26 ms, flip angle = 80 deg, voxel size = 3 × 3 × 3.5 mm³, 0 mm slice gap) oriented parallel to the AC-PC plane were acquired, covering the entire brain. For an inflated format of the cortex used for retinotopic mapping and an automated parcellation method (Freesurfer), T1-weighted MR images (MP-RAGE; 256 slices, TR = 2 s, TE = 26 ms, flip angle = 80 deg, voxel size = 1 × 1 × 1 mm³, 0 mm slice gap) were also acquired during the fMRI scans for the MVPA.

**Data availability.** All relevant data are available from the authors on request. All computer code used to generate results that are central to the paper's conclusions can be accessed following the links hereafter. For the decoding analysis, the SLR toolbox can be freely downloaded from http://www.cns.atr.jp/~oyamashi/SLR_WEB.html. For the meta-d′ analysis of behavioural data, we used the Matlab functions freely available at http://www.columbia.edu/~bsm2105/type2sdt/; (ref. 36).

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

## Acknowledgements

The study was conducted in 'Brain Machine Interface Development' under the Strategic Research Program for Brain Sciences by AMED of Japan. A.C. was supported by a Japanese Government Monbukagakusho MEXT grant. This work was supported partially by the National Institute of Neurological Disorders and Stroke of the National Institutes of Health (Grant No. R01NS088628) and the Templeton Foundation (Grant No. 21569), to H.L. The funders had no role in study design, data collection and analysis, decision to publish or preparation of the manuscript. We thank Drs Takeo Watanabe, Yuka Sasaki, Kazuhisa Shibata and Giuseppe Lisi for the very useful comments and suggestions throughout the development of the study, as well as the anonymous reviewers for greatly improving this manuscript.

## Author contributions

A.C., H.L. and M.K. designed the study while actively discussing with other co-authors; A.C., K.A. and A.K. implemented the experiment; A.C. conducted the experiment; A.C., H.L. and M.K. analysed the results with support of K.A. and A.K. Last, A.C., H.L. and M.K. wrote the manuscript.

## Additional information

**Competing financial interests:** There is a potential financial conflict of interest; one of the authors is the inventor of patents related to the neurofeedback method used in this study, and the original assignee of the patents is ATR, with which M.K. is affiliated. The remaining authors declare no competing financial interests.

