## [Peer Review File · Nature Communications]

Reviewers' comments:

Reviewer #2 (Remarks to the Author):

The authors have been highly responsive in their revisions, and the revised manuscript is strengthened in many respects leading to an overall stronger study. The inclusion of all 17 subjects for the decoding analysis is appropriate, and the more detailed analysis of the effects of training with DecNef by day provide for a more fine-grained view of its impact on confidence ratings. The modeling approach presented in Figs 6-7 seems a overly complex and parameterized in my view, but is acceptable with respect to providing a method for quantifying these effects and performing statistical analysis.

In general, the paper is methodologically rigorous but there is one remaining concern. The rectification of the leftward vs. rightward motion signals is a relevant analysis and reasonably motivated to test for effects of confidence, but applying decoding to these rectified signal seems more complex than necessary and could lack the sensitivity of a simpler analysis.

The concern about applying a decoder to the rectified signal is the following. If the initial left-right classifier is applied to noisy fMRI data and performs above chance but poorly at estimating the appropriate position of the hyperplane (e.g., biased toward the true mean of leftward or rightward motion), then the resulting rectified signal would become less precise at distinguishing different levels of sensory evidence, and this in turn could impede the ability to perform accurate binary classification of confidence.

It would be simpler to look at the linear classifier obtained from training on leftward vs. rightward motion and to report the mean distance and direction from the hyperplane for leftward motion high confidence (e.g., negative values for leftward, positive for rightward), leftward motion low confidence, rightward motion low confidence, and rightward motion high confidence. If such an analysis is performed, does one see evidence of graded effects of confidence in motion-sensitive brain areas?

Note that a positive finding would not preclude the main findings of the paper, namely that DefNef feedback based on parietal and prefrontal activity can modify confidence judgments. However, it would be relevant to the issue of whether effects of confidence appear to be fully dissociated from the strength of sensory evidence found in cortical visual areas.

Specific comments

"A central controversy in metacognition studies concerns whether confidence directly reflects the reliability of perceptual or cognitive processes, as suggested by normative models. The affirmative view enjoys popularity in the computational and animal literatures, but it has also been suggested that confidence may depend on a late-stage estimation dissociable from perceptual processes.

- not clear which is the normative/affirmative view

The writing in the introduction could be tightened and would benefit from some editing and fine-tuning. Recommend removing acronyms like e.g., and i.e., from the text and adjusting terminology such as "perception psychophysics".

Responses to Referee

First of all, we would like to thank the referee for the constructive and rigorous comments. We have carefully addressed all of the questions in this revised version of the manuscript, we hope without any exception.

Responses to Comments by Referee 2

Comment 1: In general, the paper is methodologically rigorous but there is one remaining concern. The rectification of the leftward vs. rightward motion signals is a relevant analysis and reasonably motivated to test for effects of confidence, but applying decoding to these rectified signal seems more complex than necessary and could lack the sensitivity of a simpler analysis.

The concern about applying a decoder to the rectified signal is the following. If the initial left-right classifier is applied to noisy fMRI data and performs above chance but poorly at estimating the appropriate position of the hyperplane (e.g., biased toward the true mean of leftward or rightward motion), then the resulting rectified signal would become less precise at distinguishing different levels of sensory evidence, and this in turn could impede the ability to perform accurate binary classification of confidence.

It would be simpler to look at the linear classifier obtained from training on leftward vs. rightward motion and to report the mean distance and direction from the hyperplane for leftward motion high confidence (e.g., negative values for leftward, positive for rightward), leftward motion low confidence, rightward motion low confidence, and rightward motion high confidence. If such an analysis is performed, does one see evidence of graded effects of confidence in motion-sensitive brain areas?

Note that a positive finding would not preclude the main findings of the paper, namely that DefNef feedback based on parietal and prefrontal activity can modify confidence judgments. However, it would be relevant to the issue of whether effects of confidence appear to be fully dissociated from the strength of sensory evidence found in cortical visual areas.

Answer 1: We thank the referee for raising this important point. We agree with the referee that the rectification analysis may implicitly miss possible biases in the classifiers. These biases may result in the position of the hyperplane being shifted toward the mean of either motion direction signals, impeding the proper secondary classification of confidence (high vs. low).

Reflecting the comment, we have incorporated a new analysis that takes the natural values of the linear discriminant function (LDF) with a full range from negative to positive values, pooled from all participants and sorted according to confidence levels. The LDF is computed from the SLR weights of the motion decoder; with this approach we show that LDF values do not specifically reflect the confidence level. The relationship between LDF and confidence results in a significant correlation only in area V1/V2, for negative LDF values. The referee's intuition was indeed correct, as such a result is easily dissipated with a less transparent approach. We now present this result in a new figure in the main text (new figure 6), and added the according passage (page 9, paragraph 2).

Furthermore, dividing the LDF values in large and small, for both the negative and positive parts, a significant difference in the associated confidence was found only in area V1/V2, as would be expected given the result reported above.

Thus, albeit to a limited extent in area V1/V2, even with a more sensitive approach we do not see significant evidence of graded effects of confidence in motion-sensitive areas.

As we also believe these new analyses more directly address the relation between sensory evidence and confidence, we now present the results in three new figures, one in the main text (new Fig. 6), as well as two in supplementary (Supplementary Fig. 3 and 4), for a more fine-grained view.

Furthermore, given the limitation in the number of display items (max 10 items), we have instead moved the previous figure 7 (mathematical modeling results) from the main manuscript to supplementary (Supplementary Fig. 8)

Figure 6. Relationship between linear classifier output (leftward vs. rightward motion) and confidence levels. Here we graphically assessed the relationship between confidence and the output of the classifier (Linear Discriminant Function, LDF) constructed based on the perceived direction of motion (leftward, rightward). Larger magnitude of LDF value represents trials of higher signal strength. For each ROI, black circles represent binned data points pooled from all participants, at each confidence level, respectively. The size of the circles reflect the number of data points within each bin; each side of the LDF function was subdivided into 20 bins. Thick lines (dark for negative LDF values, light for positive LDF values) are linear fits to the LDF against confidence levels. Based on normative optimality models, one would expect higher absolute LDF magnitude to be associated with higher confidence ratings, thereby forming a “v-shaped” pattern on these plots. V1/V2 alone shows a relevant significant correlation between negative LDF values (leftward motion), and confidence (Pearson’s r , corrected for multiple comparisons across ROIs). In all other ROIs, there seem to be negligible meaningful relationship between LDF magnitude and confidence. ROIs: V1/V2, V3A, hMT, FFG - fusiform gyrus, IPL - inferior parietal lobule, IFS - inferior frontal sulcus, MFS - middle frontal sulcus, MFG - middle frontal gyrus.

Supplementary Figure 3. Relationship between linear classifier output (leftward vs. rightward motion) and confidence. Here, negative and positive values of the classifier output (LDF values) corresponds to leftward and rightward motion, respectively. Reflecting the results in Fig. 6, there was no difference in confidence level as a function of the absolute classifier output (large or small) in most of the ROIs in neither direction (left or right). Only exception was V1/V2 where the magnitude of the leftward classifier output (negative LDF) was related with a significant difference in confidence. For all other ROIs, the confidence levels did not significantly differ between the two levels of LDF magnitude (large or small). The threshold to define large vs. small LDF was defined as the median at the group level for each ROI. Center values represent mean, error bars represent s.e.m. *** $P < 10^{-3}$, paired t-test between large/small LDF magnitude, corrected for multiple comparisons (two, negative/positive LDF values).

Supplementary Figure 4. Subject level analysis of the relationship between linear classifier output (leftward vs. rightward motion) and confidence. In accordance with the results at the group level (Fig. 6), there was no significant correlation between negative LDF and confidence levels in most of the ROIs, with only one exception in V1/V2 (corrected for multiple comparisons across participants). For each participant two linear fits were plotted, one for leftward motion (negative LDF) and the other for rightward motion (positive LDF).

Specific comments:

"A central controversy in metacognition studies concerns whether confidence directly reflects the reliability of perceptual or cognitive processes, as suggested by normative models. The affirmative view enjoys popularity in the computational and animal literatures, but it has also been suggested that confidence may depend on a late-stage estimation dissociable from perceptual processes.

- not clear which is the normative/affirmative view

The writing in the introduction could be tightened and would benefit from some editing and fine-tuning. Recommend removing acronyms like e.g., and i.e., from the text and adjusting terminology such as "perception psychophysics".

Answer. We thank the referee for pointing out these ambiguities in the writing. We have modified the first sentence of the abstract in order to make the statement clearer.

"A central controversy in metacognition studies concerns whether subjective confidence directly reflects the reliability of perceptual or cognitive processes, as suggested by normative models based on the assumption that neural computations are generally optimal."

As well, we have modified other relevant part of the abstract and the introduction according to the referee's suggestions, for a more eloquent readability.

REVIEWERS' COMMENTS:

Reviewer #2 (Remarks to the Author):

The revised manuscript addresses my concerns. The additional control analyses are appropriate. The revised intro is easier to understand. Overall, this is a very nice study.